# Biological rhythms in the deep-sea hydrothermal mussel *Bathymodiolus azoricus*

Audrey M. Mat [1,2✉], Jozée Sarrazin [2], Gabriel V. Markov [3], Vincent Apremont[1,2], Christine Dubreuil[1], Camille Eché[4], Caroline Fabioux [1], Christophe Klopp [5], Pierre-Marie Sarradin [2], Arnaud Tanguy[6], Arnaud Huvet [1] & Marjolaine Matabos [2✉]

Biological rhythms are a fundamental property of life. The deep ocean covers 66% of our planet surface and is one of the largest biomes. The deep sea has long been considered as an arrhythmic environment because sunlight is totally absent below 1,000 m depth. In the present study, we have sequenced the temporal transcriptomes of a deep-sea species, the ecosystem-structuring vent mussel *Bathymodiolus azoricus*. We reveal that tidal cycles predominate in the transcriptome and physiology of mussels fixed directly at hydrothermal vents at 1,688 m depth at the Mid-Atlantic Ridge, whereas daily cycles prevail in mussels sampled after laboratory acclimation. We identify *B. azoricus* canonical circadian clock genes, and show that oscillations observed in deep-sea mussels could be either a direct response to environmental stimulus, or be driven endogenously by one or more biological clocks. This work generates in situ insights into temporal organisation in a deep-sea organism.

[1] Univ Brest, Ifremer, CNRS, IRD, LEMAR, F-29280 Plouzané, France. [2] Ifremer, EEP, F-29280 Plouzané, France. [3] Sorbonne Université, CNRS, Integrative Biology of Marine Models (LBI2M), Station Biologique de Roscoff (SBR), 29680 Roscoff, France. [4] GeT-PlaGe, Genotoul, INRA Auzeville, Auzeville, France. [5] INRA, Plate-forme Genotoul Bioinfo, UR875, Auzeville, France. [6] Sorbonne Université, CNRS, Lab. Adaptation et Diversité en Milieu Marin, Team ABICE, Station Biologique de Roscoff, 29680 Roscoff, France. ✉email: audrey.mat@hotmail.com; marjolaine.matabos@ifremer.fr

Relentless environmental cycles have favoured the evolution of biological clocks in terrestrial and coastal organisms. These oscillators temporally shape living systems from the molecular to the organismal level, generating biological rhythms that are ubiquitous across taxa[1]. Whereas the circadian clock (~24-h) has been molecularly characterised, it is not the only timekeeping system that nature provides. Marine ecosystems are shaped by environmental cycles ranging from a few hours (tidal and solar cycles) to a year (seasons)[2]. Biological rhythms related to all of these environmental cycles have been described[2,3]; however the clock(s) that drive these rhythms remain enigmatic. In aperiodic environments, organisms may or may not show biological rhythms. Indeed, subterranean animals such as mole-rats, that rarely see daylight, exhibit a clear circadian rhythm of locomotor activity[4], supporting the hypothesis that internal timing is probably adaptive[1]. Inversely, the continuous lighting conditions of summer and winter at high latitudes induce a seasonal attenuation in daily rhythmic activity in reindeers living far above the Arctic circle, suggesting that a reduced circadian system may be of ecological value for animals living in seasonally arrhythmic environments[5].

The deep sea (>200 m depth) represents ~93% of the biosphere in volume and is the Earth's greatest unexplored ecological research frontier. Very little is known about the environmental cycles that influence deep-sea ecosystems. Due to darkness, mean temperature < 4 °C, high pressures, and limited food availability, the deep sea has long been considered as an environment with low and steady biological rates[6,7]. The discovery of hydrothermal vents (1977) changed our perspective because these highly productive ecosystems represent ephemeral biomass hotspots, fuelled by chemical energy. Since then, the temporal structure of vent ecosystems has been investigated through the prism of transition events, fluctuating hydrothermal activity, or environmental disturbances caused by tectonic or volcanic activity[8]. Vents are actually not as unstable as previously considered: the faunal communities from edifices in the Mid-Atlantic Ridge (MAR) and in the South Pacific were shown to be stable on a decadal scale, although variations on shorter time-scales may occur[9,10]. However, besides outflow variations, the influence of time on vents is an open question. While totally blind to light:dark cycles (L:D), the deep ocean is not an aperiodic zone: internal tides occur at all depths worldwide, resulting from the interaction of the barotropic tide with underwater topography[11]. In hydrothermal vents, where hot, anoxic fluids are discharged into the ocean, the fauna relies on local production by chemoautotrophic microorganisms that use the fluids as a source of energy[12]. Here, tides strongly affect local abiotic factors such as temperature and currents[13] and are thus more likely to affect fauna. Indeed, latest imagery studies indicate that tides influence the abundance of hydrothermal species[14–16] and growth of *Bathymodiolus thermophilus* mussels[17] below 2000 m in the Pacific Ocean. We therefore hypothesised the existence of biological rhythms in hydrothermal species, which has not yet been investigated due to technical constraints. To date, a circadian clock has only been shown in a deep-sea species by exposing *Nephrops norvegicus* living naturally between 20 and 500 m depth to experimental L:D cycles and constant conditions in the laboratory[18]; clock genes candidates were identified[19]. Feeding behaviour and melatonin secretion were also suggested to be potentially rhythmic in fish caught from trawls below 1000 m depth[20,21], and circadian clock gene candidates were identified in the hydrothermal vent crab *Austinograea alayseae*[22]. In addition, there is also evidence of lunar rhythms and seasonal cycles in the reproduction of deep-water invertebrates in the field and in the laboratory[23].

The present project was designed to explore biological rhythms and clocks in the vent mussel *Bathymodiolus azoricus* directly on the seafloor (i.e., in situ). This dominant engineer species can represent up to 90% of the biomass on MAR edifices[24]. *Bathymodiolus azoricus* hosts sulfur- (SOX) and methane-oxidizing (MOX) symbiotic bacteria in its enlarged gills[25]. Vent mussels are also a key taxa because they belong to the Mollusca, one of the most species-rich marine phylum[26]. Due to the extremely challenging nature of applying temporal protocols in the deep sea, the starting points were to gather preliminary data supporting the hypothesis of biological rhythms in *B. azoricus*, and to design and test a protocol to sample and fix mussels in a way that would be compatible with in situ chronobiological research.

Our work reveals that the valve behaviour and transcriptome of *B. azoricus* show rhythmic patterns. Tidal cycles dominate the transcriptome of mussels living on the MAR at 1688 m, where environmental data also exhibit a tidal signal. Daily cycles dominate the transcriptome of mussels exposed in the laboratory to 12:12 L:D cycles, suggesting that deep-sea mussels could perceive light. Although there is no clear oscillation of clock transcripts, *B. azoricus* possesses the canonical circadian clock genes. Circadian transcripts in situ and circatidal transcripts in the laboratory suggest that biological rhythms in *B. azoricus* could be driven by biological clock(s).

## Results

**Bathymodiolus azoricus behaviour on the MAR.** The project was focused on the Lucky Strike (LS, 1688 m depth) and Menez Gwen (MG, 834 m depth) vent fields on the MAR (Fig. 1a). Both fields harbour several active vent sites colonised by dense assemblages of *B. azoricus*. Since they are located in an area designated for long-term monitoring efforts, their abiotic and biotic environments have been extensively studied. This is particularly true for LS which has been part of the European program EMSO (European Multidisciplinary Seafloor and water-column Observatory) since 2010 and hosts the EMSO-Azores deep-sea observatory. The literature on the influence of environmental cycles on vent fauna is scarce and mostly based on discrete video-recording used to monitor variations in the abundance of vent species in the Atlantic and Pacific[14–16]. The identification of significant tidal signals in vent species inhabiting the Pacific[14,15,17] motivated the question of a potential rhythmic activity in Atlantic mussels at the organism level. Before deploying significant high-sea resources, we thus analysed the valve-activity (open or close) of 31 mussels over one month on the Eiffel Tower edifice at LS using video-recordings acquired in 2014 via the observatory (Fig. 2a, b). The Dutilleul multi-frequential periodogram analysis (MFPA) revealed that 12.5 h and 26.3 h were the dominant periods in the percentage of opened mussels (*p*-value < 0.05). The Whittaker–Robinson-type periodogram also indicated significant semi-diurnal and diurnal periods (i.e., 12 h and its harmonics, *p*-value < 0.05; Fig. 2c). Despite the use of artificial white illumination and the limited sampling frequency, these behavioural data supported the hypothesis of biological rhythms in deep-sea mussels.

**Bathymodiolus azoricus gills' temporal transcriptome.** We therefore extended our analysis to *B. azoricus* transcriptome by collecting mussels at LS (1688 m depth) under red light instead of the usual bright white illumination used for deep-sea biology (see "Methods" and Fig. 1b, c). Sampling was performed every 2 h 4 min for 24 h 48 min, and mussels were fixed directly in situ.

We worked on gills, a key organ for both nutrition and respiration in marine bivalves and the first sensor of the environment[27]. In *B. azoricus*, this organ also hosts the chemoautotrophic symbionts[25]. Our reference de novo transcriptome built from 81.8 Mb of sequences comprised 34,196

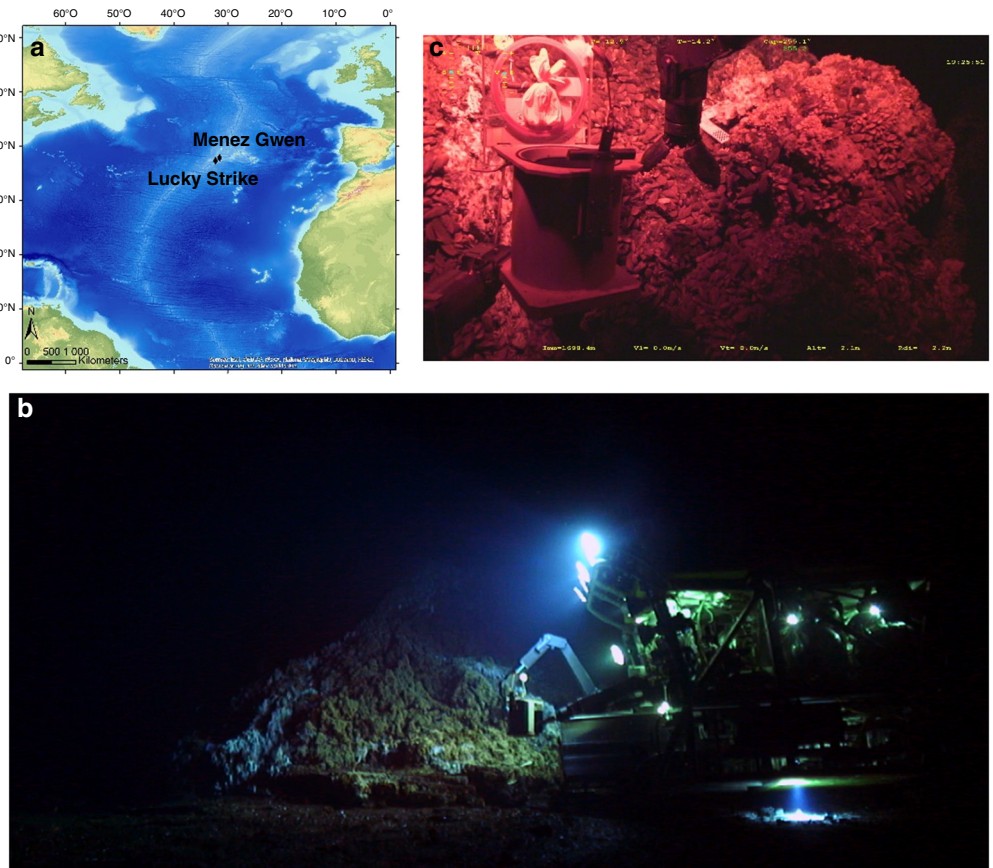

**Fig. 1 Sampling strategy to allow deep-sea chronobiology studies. a** The work was carried out on *Bathymodiolus azoricus* mussels from the Mid-Atlantic Ridge (MAR). The in situ study was performed on the Eiffel Tower edifice, at the Lucky Strike vent field (1688 m depth). For the laboratory experiment, mussels were sampled at the Menez Gwen vent field (834 m depth). Sources: Esri, GEBCO,NOAA, National Geographic, DeLorme, HERE, Geonames.org. **b** A typical work session of the Remotely Operated Vehicle (ROV) *Victor6000* in the deep sea, using artificial white light. © JY Collet - Bienvenue Productions - Ifremer. **c** The sampling strategy was designed to sample and readily stabilise mussel tissues under red light in the deep sea. MOMARSAT 2017 cruise. © Ifremer.

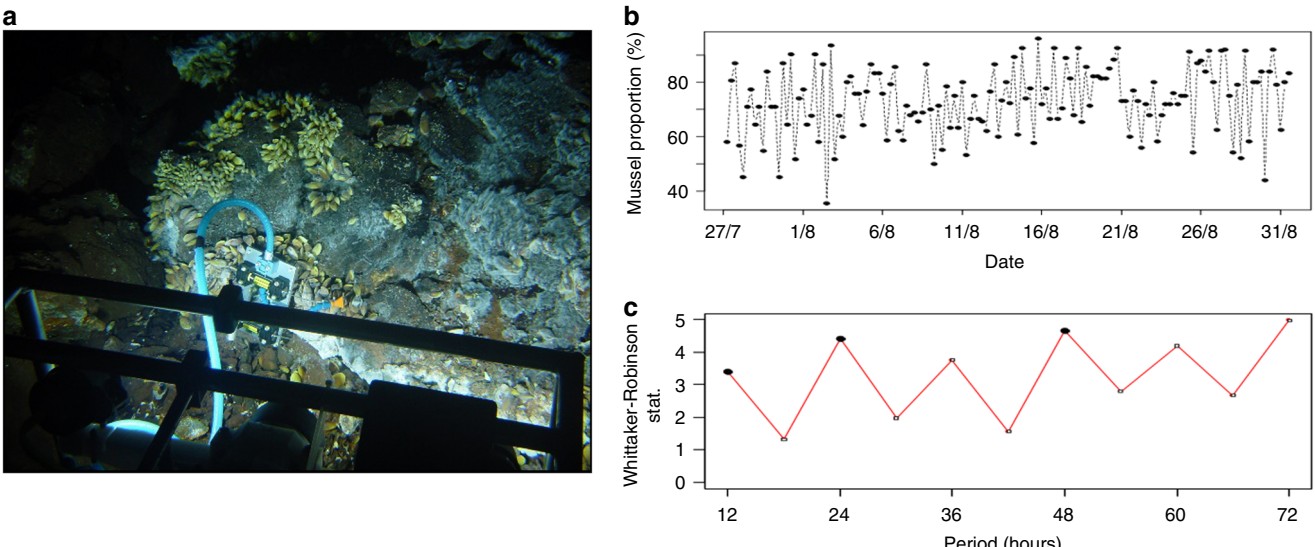

**Fig. 2 *Bathymodiolus azoricus* valve opening was rhythmic on the Mid-Atlantic Ridge. a** The ecological observation module TEMPO records 2-min video sequences four times a day, under white artificial light, on the Eiffel Tower vent edifice, Lucky Strike vent field (1688 m depth). **b** Proportion of *B. azoricus* mussels ($n = 31$) that were open at 0, 6, 12, and 18 h UTC between 27 July and 31 August 2014. **c** Whittaker–Robinson periodogram analysis of mussels' valve behaviour. Black dots indicate significant periods.

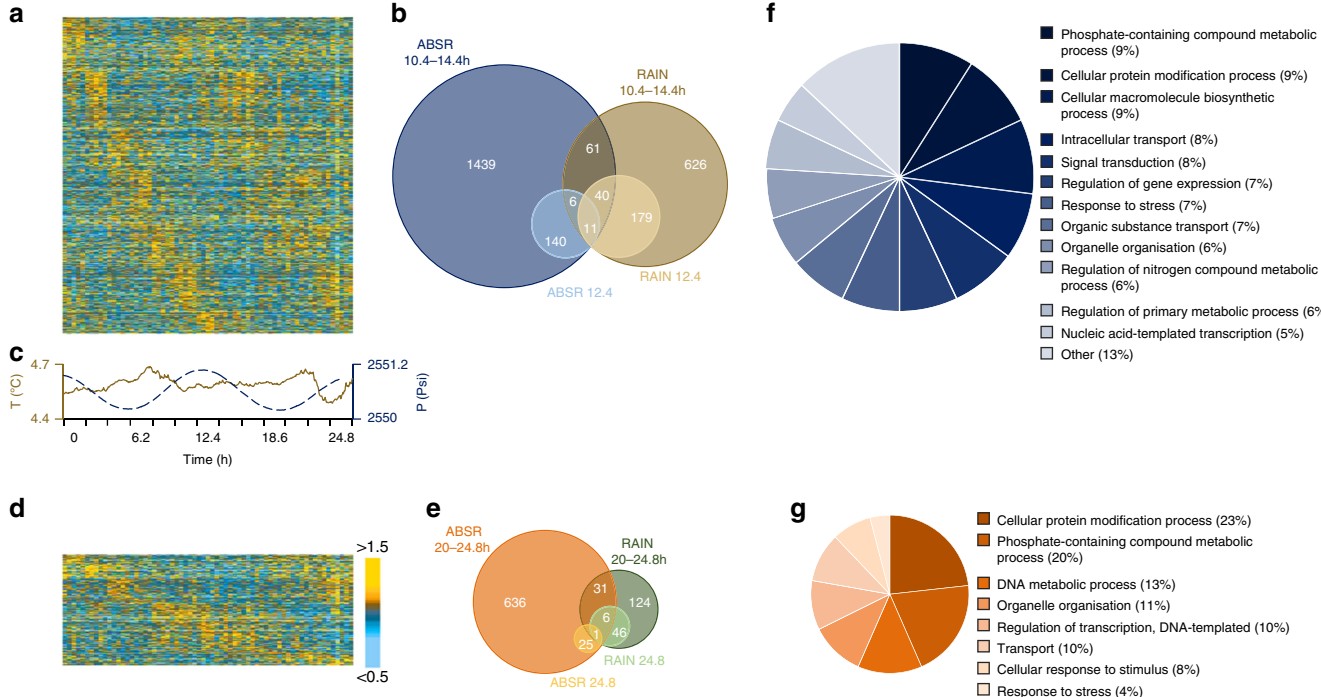

**Fig. 3 *Bathymodiolus azoricus* gill transcriptome was cyclic at the hydrothermal vent field (1688 m depth).** Normalisation performed with down-sampling and DESeq2. **a** Heatmap of median-normalised expression patterns of all 2502 rhythmic transcripts detected in the interval 10.4–14.4 h. **b** Euler diagrams detailing the number of rhythmic transcripts detected in the interval 10.4–14.4 h and specifically with a period of 12.4 h using both RAIN (dark and light brown disks) and ABSR (dark and light blue disks) methods. **c** Seawater temperature (brown continuous line) and pressure (blue dotted line) profiles recorded on the Lucky Strike vent field at the time of sampling. **d** Heatmap of median-normalised expression patterns of all 869 rhythmic transcripts detected in the interval 20–24.8 h. **e** Euler diagrams showing the number of rhythmic transcripts detected in the interval 20–24.8 h and specifically with a period of 24.8 h using both RAIN (dark and light green disks) and ABSR (dark and light orange disks) methods. **f** GO terms associated to all the rhythmic transcripts detected in the range 10.4–14.4 h and **g**, in the range 20–24.8 h. All heatmaps are single-plotted and represent five individuals per time point. Rhythmic transcripts are ordered by phase. Heatmap colours: median-normalised expression levels greater than 1.5-fold are shown as gold yellow; expression levels less than 0.5-fold are shown as light blue. Heatmap heights and disk areas are proportional to the number of transcripts.

contigs, with a weighted median N50 of 3310 bases. The percentage of contigs longer than 1 kb was 77.1%. AT content was 66.8%, which is similar to other marine mollusc genomes, e.g., *Pinctada fucata*[28] or *Lottia gigantea*[29]. The BUSCO score of this transcriptome was 94.2% in total, with 88.4% and 5.8% for single- and double-copies, respectively, attesting the assembly completeness[30]. The alignment rates to the reference transcriptome were >96% for all samples. The annotation rate of the reference transcriptome was 47%, with 79.2% of the annotated contigs matching marine molluscan species proteins, belonging namely to the bivalves *Mizuhopecten yessoensis*, *Crassostrea gigas*, *Crassostrea virginica*, and the gastropod *Lottia gigantea*, in decreasing order of importance. Only 52 and 66 contigs matched *Bathymodiolus* sp. SOX and MOX symbionts[31,32], indicating that we indeed were able to isolate the mussel transcriptome from its symbionts.

**Tidal cycles dominate *B. azoricus'* transcriptome in situ.** We used two different normalisation approaches (see "Methods"), one by down-sampling the data as validated for the oyster *C. gigas*[33], and the other by also using DESeq2 as performed in the sea anemone *Aiptasia diaphana*[34]. Both approaches gave similar results in terms of dominant cycles and number of rhythmic transcripts. We present the results obtained with the DeSeq2 normalisation here while the results gained with the down-sampling procedure are given in the Supplementary Fig. 1.

Putative cycling transcripts were explored using two methods that can detect nonsymmetric wave forms in short datasets, the non-parametric method RAIN[35] and the Bayesian algorithm

ABSR[36]. In situ, 7.4% of the transcriptome (i.e., 2502 transcripts) oscillated in the range 10.4–14.4 h, and 1.1 % of the transcriptome (376 transcripts) showed a period of 12.4 h (Fig. 3a, b). The sampling site was clearly under tidal influence during the sampling period (Fig. 3c) with both temperature and pressure (range 2550.2–2551.1 psi) oscillating with a ~tidal period (12.3 and 12.7 h; Dutilleul MFPA, both *p*-values < 0.05). While the temperature range was small, 4.5–4.7 °C, concentrations in sulphides, methane and copper can vary with one order of magnitude between 4.0 and 4.5 °C on the LS vent field[37,38]. These changes have profound implications for the vent fauna in terms of physiological tolerance. In addition, 2.6% of the transcriptome (869 transcripts) oscillated in the range 20–24.8 h, with 0.2% of the transcripts (78 transcripts) having a period of 24.8 h (Fig. 3d, e). The list of the most robust rhythmic transcripts is provided in Supplementary Data 1. A significant part of the transcription thus appeared rhythmic in *B. azoricus*.

In the absence of internal control, i.e., a rhythmic gene whose expression is cyclic and whose profile is known, the character-istics of the rhythmic transcripts were compared with those of all the transcripts in the reference transcriptome to analyse whether the different subsamples were representative. Transcripts with low counts usually exhibit more noise[39], and present therefore an increased risk of false positives. For the in situ experiment, comparing the transcripts abundance revealed that the med-ian normalised counts for the reference, the tidal transcripts, and the daily transcripts were 298, 308, and 330, respectively. The percentages of annotated transcripts were also compared. In the reference transcriptome, 47.2% of the contigs were annotated. In

the in situ experiment, 49.0% of the tidal transcripts and 44.4% of the daily transcripts were annotated. Globally, the different subsamples of rhythmic transcripts shared similar characteristics with the reference transcriptome in terms of both quantification and annotation, indicating that rhythmic analyses did not isolate subsamples of a lower quality.

While ~47% of the transcriptome was annotated to proteins, only 31% was associated with gene ontology terms (GO), giving a limited understanding of the biological functions shaped by environmental cycles. In situ, the analyses of the biological processes showed that tidal oscillations globally influenced many cellular processes such as transport, cellular organisation, localisation and communication; and metabolic processes related to organic substances, primary metabolites and nitrogen compounds (Fig. 3f). The enrichment analysis showed the importance of responses to environmental stresses with several entries related to reactive oxygen species (Supplementary Data 2a). As oxygen, iron, and sulphide at hydrothermal vents can lead to the production of reactive oxygen species[40,41], this is likely related to the oxidative stress that hydrothermal organisms have to face. The daily transcripts in situ were also related to transport, cellular organisation and metabolic processes (Fig. 3g), and interestingly there was also a major contribution of responses to stimulus and stress. The enrichment analysis showed that the first 17 GO entries (out of 101) are related to the c-Jun N-terminal kinase (JNK) and the mitogen-activated protein kinase (MAPK) cascades, the NIF/NF-κB cascade and stress-activated kinases (Supplementary Data 2b). The NIF/NF-κB signalling pathway is a pleiotropic module involved in many biological processes including immunity, apoptosis, inflammation, and is a key responder to changes in the environment[42]. The JNK signalling pathway, which is part of the MAPK one, is involved in immunity, inflammation, apoptosis and metabolism as well. The JNK cascade is activated by environmental stresses such as thermal and oxidative stress, ionising radiation and DNA damage[43,44], all of which are characteristic of the abiotic conditions found at hydrothermal vents. Interestingly, three contigs annotated as belonging to Bathymodiolus sp. symbionts (two MOX and one SOX) were present in the list of tidal transcripts and two (one MOX and one SOX) in the list of daily transcripts. As symbiont transcripts are very rare in our reference transcriptome, the presence of rhythmic bacterial transcripts raises questions about the relationship between the mussel and the bacteria in the rhythmic functioning of the holobiont, a topical question in other symbiotic species such as the sea anemone Aiptasia diaphana[34].

**Daily cycles dominate B. azoricus' transcriptome in the lab.** Because a progressive diversification from shallow- to deep-water species has been suggested for vent species[45], we also investigated in parallel mussels' physiology in the lab under a 12 h:12 h L:D schedule to determine whether B. azoricus could still respond to light. We used organisms from the nearby MG vent field (834 m depth) that can be studied at atmospheric pressure[25], unlike LS mussels. In the laboratory, under L:D entrainment (Supplementary Fig. 2c), daily oscillations became more prominent than tidal ones, with 7.5 % of the transcriptome (2511 transcripts) oscillating in the daily range (20–24.8 h), compared to 5.9% in the tidal range (1988 transcripts). Similarly, 2.8% of the transcriptome (947 transcripts) expressed a period of 24.8 h and 0.6% (209 transcripts) a period of 12.4 h (Supplementary Fig. 2a–b, d–e). Again, rhythmic transcripts shared similar characteristics with the reference transcriptome in terms of both quantification and annotation. The median normalised counts for the reference, the tidal transcripts, and the daily transcripts were 227, 233, and

246, respectively. The tidal transcripts and the daily transcripts were 48.8% and 46.8% to be annotated, compared with 47.2% of the contigs in the reference transcriptome.

As in the in situ experiment, tidal transcripts were related to cellular and metabolic processes (Supplementary Fig. 2f). In addition, 12% of the GO entries in the enrichment analysis were related to responses to stimuli (DNA damages, endogenous, biotic and chemical stimuli among others; Supplementary Data 2c). Daily transcripts covered a range of cellular and metabolic processes (Supplementary Fig. 2g); gene expression GO terms were well represented, indicating that biological cycles can influence many physiological functions (Supplementary Data 2d). Again, two contigs annotated to Bathymodiolus sp. symbionts were present in the list of daily transcripts but none in the list of tidal transcripts.

For the transcripts that were rhythmic in the range 10.4–14.4 h, 152 transcripts were rhythmic under both in situ (2502 rhythmic transcripts) and laboratory conditions (1988 rhythmic transcripts). For the transcripts that were rhythmic in the range 20–24.8 h, 63 transcripts were rhythmic under both in situ (869 rhythmic transcripts) and laboratory conditions (2511 rhythmic transcripts). These transcripts are detailed in Supplementary Data 3a, b.

While exposing B. azoricus to light is ecologically unrealistic, it provides two crucial pieces of information. First, B. azoricus transcriptome oscillations were different in situ and under LD 12:12, indicating that there was a likely biological response to light in these deep-sea mussels. Second, we identified transcripts with a circadian rhythmicity in situ, without known daily cycle, and transcripts with a circatidal rhythmicity in the laboratory, without tidal signal. It suggests that both tidal and daily rhythms could be endogenously generated in B. azoricus.

**Bathymodiolus azoricus canonical circadian clock genes.** We identified B. azoricus homologs for Clock, bmal, period, timeless, timeout, light receptive-type cryptochromes (cry), and transcriptional repressor-type cryptochrome (Fig. 4). Clock, bmal, and period are core components of the circadian clock in both vertebrates and flies[1]. One Clock transcript was identified (2780 nt/926 aa; Supplementary Fig. 3), with a second isoform (2575 nt/858 aa). Two bmal sequences were identified (609 nt/202 aa and 3392 nt/1130 aa; Supplementary Fig. 3) and are likely part of the same transcript, but with a non-overlapping area. These two contigs were mapped to C. gigas bmal and assembled into one contig. The Clock and bmal transcripts present one bHLH (basic Helix-Loop-Helix) and two PAS (Per-Arnt-Sim) domains. One period homolog was identified (5943 nt/1980 aa; Supplementary Fig. 4), as well as one isoform (5965 nt/1988 aa). The period sequence presents one PAS and one Period_C domain. Bathymodiolus azoricus Period_C domain detected with Pfam is longer (137 aa) than those of M. yessoensis (58 aa), C. gigas (70 aa), and C. virginica (71 aa). Timeless is a core component of the circadian clock in flies but not in vertebrates, while timeout plays a role in the entrainment of the circadian clock in insects[1]. One timeless (3506 nt/1,168 aa) and two timeout (5432 nt/1810 aa and 3875 nt/1290 aa, most likely isoforms) sequences were identified (Supplementary Fig. 5). Finally, cry is a core component of the circadian clock in vertebrates and some invertebrates (the monarch butterfly) while it is part of the circadian entrainment pathway in several other insects[1]. Five sequences were identified in the cry/photolyase family (Supplementary Fig. 6): one transcriptional repressor-type cry (1974 nt/657 aa), two light receptive-type cry (3259 nt/1086 aa and 1266 nt/421 aa), one photolyase (1746 nt/581 aa), and one cry-DASH (1443 nt/480 aa). For all phylogenies, B. azoricus putative clock genes were closely related to marine

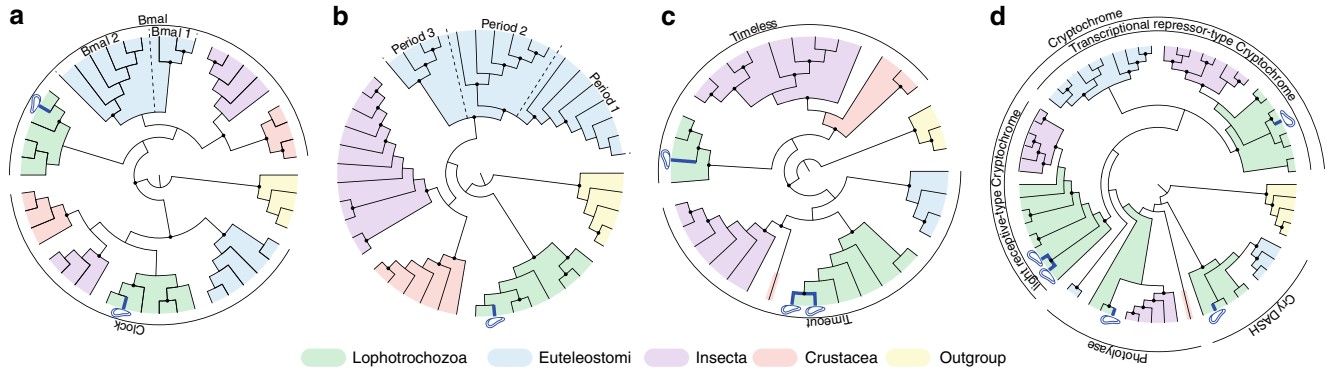

**Fig. 4 Simplified Maximum-likelihood phylogenies of the canonical circadian clock genes. a** CLOCK and BMAL/CYCLE proteins. **b** PERIOD proteins. **c** TIMELESS and TIMEOUT proteins. **d** CRYPTOCHROME and PHOTOLYASE proteins. Dots on branch indicate that the node supports a value ≥0.97 (aLRT). *Bathymodiolus azoricus* proteins are highlighted in dark blue. Green: Lophotrochozoa; light blue: Euteleostomi; purple: Insecta; red: Crustacea; yellow: outgroup.

bivalves, either the oyster *C. gigas* or the scallop *M. yessoensis* (nodes > 0.97, aLRT branch support values).

The circadian clock is based on an auto-regulatory transcription and translation feedback loop (TTFL) of the above mentioned genes[1]. We cannot establish if the TTFL is still functional in *B. azoricus* and, if positive, if it works under the vertebrate or the insect model. First, there was no global coherent rhythmic profile for all canonical circadian genes, neither in situ nor in the lab under L:D entrainment. The canonical circadian clock genes were not really cycling, although *period*, *timeless*, *photolyase*, and *cry1* showed a bimodal profile and/or were close to the *p*-value = 0.05 threshold in situ. *Period* was actually in the circatidal range in the laboratory (Fig. 5). Second, all circadian clock candidates were lowly expressed and with an important inter-individual variability: they were expressed at a level below the median normalised counts for the reference (298 and 227 for the in situ and L:D experiment, respectively), and below the quality thresholds for proper quantification (see "Methods"). *Bathymodiolus azoricus* inhabits an environment without light and was acclimated to L:D cycles in the lab for 48 h before sampling. The clock could need a longer time period to re-entrain to L:D cycles and to show significant levels of gene expression. Finally, the oscillation of clock genes is tissue-specific[46]; tissues other than gills might be more appropriate to detect clock gene oscillations in mussels and determine whether it still possesses a functional circadian clock. Nevertheless, as there were no stop codons in these sequences, the circadian clock genes are expected to encode functional proteins.

## Discussion

Biological rhythms observed directly in the deep sea is groundbreaking. *Bathymodiolus azoricus* in situ and laboratory high-resolution temporal transcriptomes provide additional resources to tackle pending questions in deep-sea biology and chronobiology, and globally enlighten our understanding of how marine organisms cope with their complex oscillatory environment.

This study establishes that tidal and daily molecular and behavioural oscillations occur in vent mussels, showing that life is not aperiodic in the dark depths where tides are an essential driver[13]. In situ pressure and temperature data recorded at LS at the time of sampling confirmed the prominent role of tides on environmental variability, as previously shown in both focus flow and mussel habitats at the same site[13,16]. At vents, tide-related variability results from oceanic tidal pressure and modulation of horizontal bottom currents[13], which in turn affects the intensity and influence of hydrothermal fluids on species. Previous

ecological studies based on video analyses suggested that vent species respond to these habitat modifications by adjusting their behaviour to ensure optimal living conditions[14–16]. This hypothesis is supported by the importance of GO terms related to environmental stresses in the present work. We suggest that *B. azoricus* cyclic activity could be interconnected with its symbionts' needs for sulphide and methane from the vent fluids in an oscillating environment. A future challenge will be to perform a triple transcriptome including *B. azoricus* and its SOX and MOX symbionts to understand how this successful symbiosis operates in time. As reported in the present work in situ, oscillations in GO biological processes related to the molecular JUN, MAPK, and NF-κB pathways have been highlighted similarly in shallow-water mussels exposed to simulated tidal and L:D cycles in the lab[47]. Oscillations in processes related to phosphate ion transport, DNA damage and repair, and response to stimulus have been reported in intertidal mussels in the field[48] and were also observed here in deep-sea mussels in the lab. This indicates that there is a common set of processes that can be rhythmic in both deep- and shallow-water mussels, depending on the environmental conditions.

The biological rhythm perspective in situ could fundamentally transform deep-sea science as well as sampling and management strategies. Organisms' internal time is currently an open question and rarely taken into account in deep-sea ecology because we know so little about temporal organisation in these environments and because of technical constraints. Including biological rhythms and temporal niches into future research protocols and environmental protection strategies could fundamentally transform our understanding of ecosystem functioning and biodiversity patterns. The present work also shows that daily rhythms gained importance in the laboratory under L:D entrainment. While this observation is likely not representative of mussels' physiology in the deep sea, it suggests that *B. azoricus* could perceive light, a trait potentially conserved from a shallow-water ancestor[45]. It should be noted that while deep-sea hydrothermal vents are blind to the L:D solar cycle, they emit low levels of thermal and non-thermal radiation[49]. Conducting responsible science at hydrothermal vents is a paramount issue, yet the largely unexplored idea that illumination during sampling may be intrusive[50] has now to be discussed based on the present data.

Our work allowed deep-sea molecular study of biological rhythms in a non-model organism while directly fixing specimens in situ. We have developed a cutting-edge protocol to work both under realistic ecological conditions and follow at best latest guidelines for omics analyses in chronobiology[51]. However, several elements must be debated. First, there is potentially a clock in

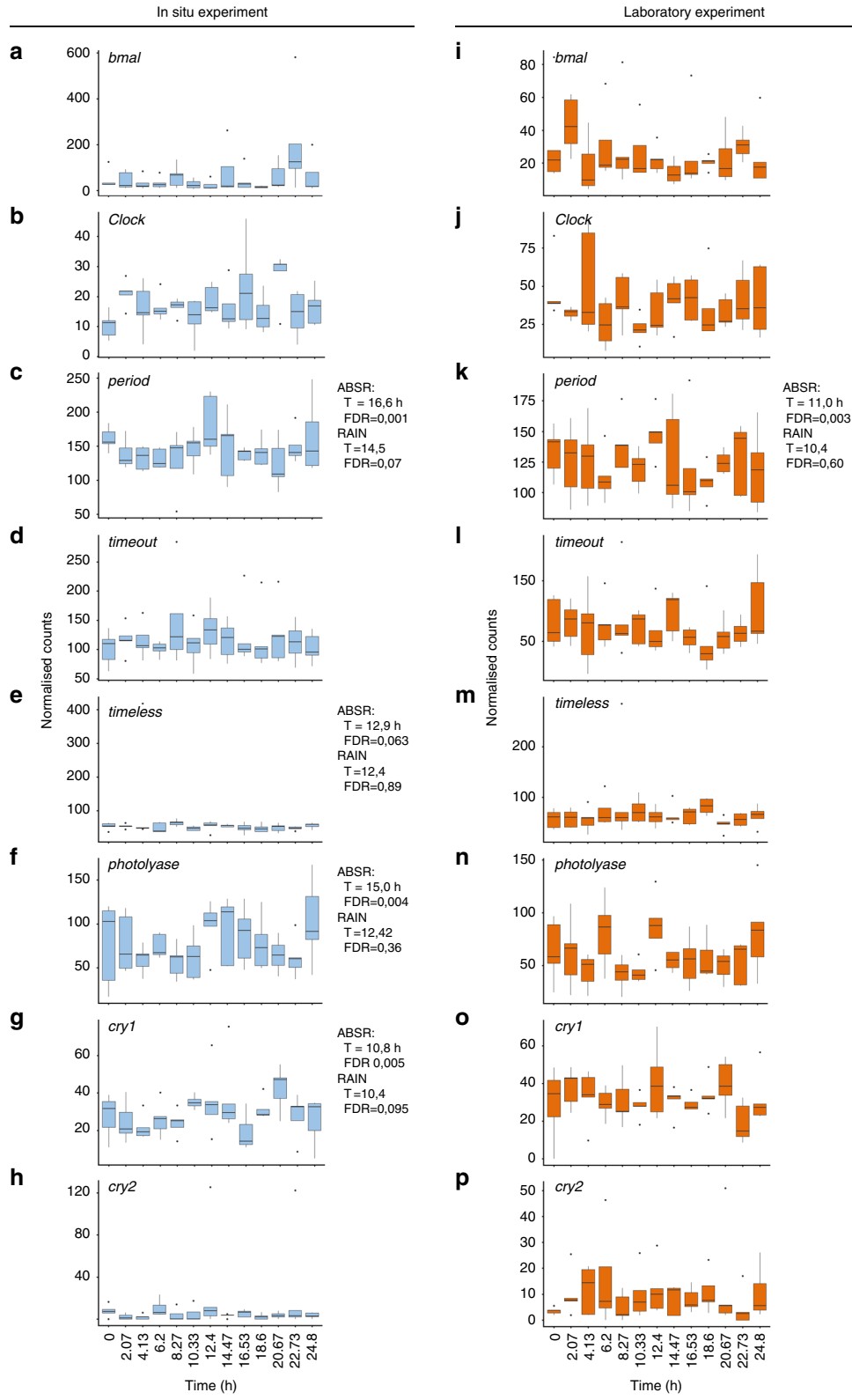

**Fig. 5 Temporal expression profiles of *Bathymodiolus azoricus* canonical circadian clock transcripts.** Expression at the hydrothermal vent (**a**–**h**, blue plots) and in the laboratory under L:D 12:12 (**i**–**p**, orange plots). Transcriptomic data, $n = 5$ mussels/time point. **a**, **i**, *bmal*. **b**, **j** *Clock*. **c**, **k**, *period*. **d**, **l**, *timeout* (MN611451). **e**, **m**, *timeless*. **f**, **n**, *photolyase*. **g**, **o**, *cry1* (MN611453). **h**, **p**, *cry2*. RAIN and ABSR results are shown when one of the analyses returned significant results or results very close to the significant threshold (FDR ≤ 0.05). Whisker plot central line: median; lower and upper hinges: first and third quartiles; whiskers: 1.5× interquartile range; points: outliers.

every cell and tissue[52], and almost 50% of all genes in the mouse genome show circadian rhythms in transcription in at least one organ[53]. The proportion of cyclic transcripts in *B. azoricus* gills, 10% total in situ and 13.4% in the lab, appeared below this value, or below the 23% of cyclic transcripts in sea anemone *Aiptasia diaphana* individuals[34]. It is however close to the value found for the oyster *Crassostrea gigas* in which 6% of the gills' transcriptome exhibited circadian expression[33]. Our data clearly show that biological rhythms influence mussels' physiology. However, capturing the full impact of rhythms on their transcriptome is another challenge that will require analysing every tissue. Second, the study was performed over 24.8 h. The consistency of daily transcripts would be of further interest over a longer sampling period. It has been shown, with the JTK_Cycle algorithm, that the detection of circadian transcripts depends on sampling resolution and read depth[54]. Sampling 12 time points over 2 full cycles is therefore recommended when possible[51]. In addition, the number of rhythmic transcripts does not reach a plateau with increasing read depths[54], meaning that no study can currently pretend to detect them all. The present read depth is in the upper range of the current recommendations for omics biological rhythm works[51]. Lastly, we used two normalisation methods and two rhythm detection methods to greatly support the consistency of our data.

The endogenous or exogenous nature of the tidal and daily rhythms reported here remains to be elucidated. Observing oscillations in an organism under driven conditions does not mean that they represent true biological rhythms generated by (an) internal clock(s). It could reflect a direct response to an environmental signal[1]. However, circadian transcripts in situ and circatidal transcripts in the laboratory suggest the presence of free-running clocks. *Bathymodiolus azoricus* possesses the canonical circadian clock genes, although they do not exhibit clear oscillations. Whether the TTFL is functional in *B. azoricus* has to be determined. In addition, TTFL-independent circadian clocks, whose molecular mechanisms are unknown, have already been reported in eukaryotes[55,56], offering future investigations in a large variety of organisms.

Knowledge on molecular biological clock(s) in marine species is severely lacking and only a few studies exist compared to data on terrestrial organisms. Separate circadian and circatidal clocks appear to operate in the crustacean *Eurydice pulchra*, but both systems appear to share common regulators[57]. Similarly, the marine worm *Platynereis dumerilii* holds separate circadian and circalunar clocks, with a modulation of the circadian rhythm by the circalunar clock[58]. On the other hand, several studies support the growing idea that canonical circadian markers are actually not intrinsically circadian, but that their oscillation depends on the dominant cycle of an organism[59–61]. The challenge is now to identify molecular tidal markers and common regulators to understand whether the circatidal and circadian clocks can fit into one conceptual framework. Although different species might have evolved to produce different timekeeping systems[3], this challenge appears particularly relevant in mussels. In littoral species, under laboratory or natural tidal and daily L:D cycles, both tidal and daily oscillations have been reported in *Mytilus edulis*[62] and *Mytilus galloprovincialis*[63] behaviour, and in *Mytilus californianus* transcriptome[47], metabolism, and cardiac activity[64]. Despite the influence of tides on mussel physiology, circadian rhythms predominate[47,63]. Free-running circadian rhythms have been reported[65] and several putative circadian clock genes have been identified[66], but the potential circatidal clock is uncharacterised. It is not known whether this is because tidal activity is non clock-driven in mussels, or because laboratory conditions or the tissues analysed were not favourable for detecting a circatidal clock, or because tidal and daily rhythms could be generated by a single

clock. In the oyster *C. gigas*, it had been suggested that circatidal rhythms could be generated by a tidally synchronised circadian clock[67]. Recent studies reported tidal expression of circadian clock genes in the laboratory[59] and in the field[68], supporting the later hypothesis. Understanding the clockwork(s) in marine organisms is complicated by their co-occurrence and interactions. Deep-sea ecosystems, light-free but exposed to tidal stimuli could therefore provide a unique opportunity for untangling the evolution of biological clocks. Even if the present work did not determine whether mussels' rhythms were free-running in constant conditions, these organisms physiology is clearly partially cyclic. Hydrothermal vents are usually considered either as modern analogues of the primitive ocean[69] or as some of the earliest habitable environments on Earth[70]. Molluscs have successfully adapted to aquatic and terrestrial habitats and are vital for many ecosystems. Bivalves, more specifically, have been present on Earth for more than 400 million years. Turning the (red) spotlight on deep-sea mussels may be central to understanding the evolution of biological timing.

## Methods

**General guidelines.** The experimental procedures comply with French law and Ifremer institutional guidelines. No statistical method was used to predetermine sample size, which was chosen as a compromise between deep-sea technical constraints and number of replicates. In this manuscript, in situ refers to samples gathered and fixed directly on the seafloor, here at deep-sea hydrothermal vent, to allow a realistic environmental approach.

**Behavioural analysis of *Bathymodiolus azoricus* in situ.** Behavioural data were recorded with the ecological observation module TEMPO, which is part of the EMSO-Azores observatory and allows the direct observation of a *Bathymodiolus azoricus* mussel assemblage at the base of the Eiffel Tower edifice (LS vent field, MAR). The module is built around an aluminium frame equipped with a handle and two adjustable feet that hosts an autonomous video camera (720 × 576 pixels) and two 35 W white LED projectors protected by an anti-fouling system for light[71]. The data analysed in this study were collected from 27 July 2014 to 31 August 2014, during which the camera recorded 2-min video sequences four times a day (i.e., 0, 6, 12, and 18 h UTC). Thirty one mussels could be tracked over the entire period. For each video, a snapshot extracted at the 25th second was analysed to determine if each individual mussel was open (coded as 1) or closed (coded by 0) using Image J©[72]. The proportion of opened mussels was then calculated for each observation date and the resulting time series was submitted to a Dutilleul MFPA to identify the dominant periods in the opening and closing of mussels. This method estimates significant periodic fractional frequencies and computes the corresponding R-square of the explained variance for each corresponding period[73]. This periodogram is particularly useful when the sampling step does not allow to discover useful periods that are whole multiples of the sampling step. However, plotting the periodogram requires a very large number of fractional values on the abscissa and would be impossible to represent graphically. We thus also show the results of the Whittaker–Robinson-type periodogram that finds a fixed set of whole periods contained in the sampling frequency. The analyses were performed with the R-package adespatial (v 0.3-7).

**Environmental data.** Temperature and pressure data were recorded every 2 min by the probe SBE 53 Bottom Pressure Recorder (Sea-bird Scientific) 26 s/n located on the LS vent field at a depth of 1726 m. The probe was located at a distance of 745 m from our sampling site.

**Sampling procedure for the in situ experiment.** Temporal and chronobiological studies in the deep sea come with strong technical constraints. First, depth prevents any direct sampling. The collection of biological samples is always a technical feat that requires a research vessel and a submersible. Second, for visibility purposes, sampling is always performed under bright illumination (Fig. 1b). It is unknown whether vent species are still capable of perceiving light but if they are, working under white light is not compatible with rhythm research: it could reset the clock[74]. One therefore needs to work under red light, which is the first part of the visible spectrum to be absorbed in seawater[75]. Third, there is a time lag before samples return on board from the deep. We thus developed a unique exploratory strategy to sample and immediately stabilise mussels under red light in situ, at 1688 m depth on the seafloor. *Bathymodiolus azoricus* mussels were sampled during the MOMARSAT 2017 cruise (July 16 and 17; doi 10.17600/17000500) at the Lucky Strike vent field of the MAR, on the NE part of the Eiffel Tower edifice, at a depth of 1688 m, using the Remotely Operated Vehicle (ROV) Victor6000. The ROV light spots were equipped with red filters (LF RL019HT Fire, Lee Filters, USA) to limit any light contamination (Fig. 1c). A total of 91 *B. azoricus* mussels (68 ± 14 mm

shell length, mean ± sd) were sampled every 2 h 4 min for 24 h 48 min, covering 2 tidal cycles and a daily cycle. All mussels were collected at the same location in an area preserved from light contamination prior to sampling; the ROV carefully approached the sampling site under red light (see Supplementary Movie 1). A separate sampling box was used for each sampling time. Each box had a useful volume of ~5 L and had been fully filled on board with in-house RNA stabilising solution. The boxes were protected, in the inside, by rubber blades that hold the solution inside the box. The stabilising solution was prepared in the following proportions (US patent 8178296 B2): 40 ml 0.5 M EDTA (VWR), 25 ml 1 M sodium citrate (VWR), 700 g ammonium sulphate (Honeywell) and 935 ml of distilled water; the pH was adjusted to 5.2 with 5 M sulfuric acid (Merck). For each sampling time, 7 to 15 mussels were collected, intentionally slightly cracked open using the ROV arm, and directly placed in a box to preserve RNA immediately upon sampling (see Supplementary Movie 2). As the volume of one mussel is about 50 ml, tissues were thus placed in 5-10 volumes of RNA stabilising solution, which is required for RNA preservation. Because the shells were lightly ground, the preservation media rapidly permeated mussel tissues to stabilise and protect cellular RNA, circumventing the need to immediately process tissue samples. Once on board, mussels with unbroken shells were discarded. Sampling boxes were surfaced in two stages, respectively 5 h 36 min and 27 h 23 min after the beginning of the experiment: time lag between sampling and surfacing varied from 5 h 36 min to 21 h 13 min. In the meantime, the boxes were stored at ambient temperature (~4.4 °C) on the deep-sea floor. Aboard, mussels were dissected in a cold room at 4 °C and gill tissues were placed in individual tubes in RNA*later*® (Sigma-Aldrich), kept at 4 °C overnight then transferred to -80 °C before further processing.

**Sampling procedure for the laboratory experiment**. *Bathymodiolus azoricus* was also studied in the laboratory to determine whether it was still capable to respond to Light:Dark cycles. Mussels were sampled during the Biobaz 2017 cruise (July 26) at the White Flames area of the Menez Gwen vent field (Mid-Atlantic Ridge), at a mean depth of 834 m, using the ROV *Victor6000*. A total of 108 mussels of similar sizes were collected for the laboratory experiment. Mussels were kept on board for 40 h in natural seawater before being transferred to the Lab Horta facility of the Universidade dos Açores. Mussels were acclimated, for 48 h only due to technical constraints, in a temperature-controlled room, in an aquarium at 6.0 ± 0.5 °C (mean ± sd, *n* = 4), and pH 6.8 under a 12 h:12 h L:D schedule (one white 4 W LED, 3000–6500 K, model PU2014, Ante, Portugal). Light intensity was ~815 lux (TES 1335 Light Meter, TES Electrical Electronic Corp., Taiwan) in the air at water surface, and ~250 lux in the air at the bottom of the aquarium. Night sampling was performed under red light. A total of 65 *B. azoricus* mussels (63 ± 8 mm shell length, mean ± sd) were then sampled every 2 h 4 min for 24 h 48 min. Mussel tissues were dissected and stored in liquid N₂ before further processing.

**RNA extraction**. Total RNA were isolated using Extract-all reagent (Eurobio, France) at a concentration of 1 ml·30 mg⁻¹ tissue, treated with DNAse I (DNAse Max Kit, Qiagen, Germany) and assayed for concentration and quality. All samples complied with purity criteria (OD260/OD280 and OD260/OD230 > 1.8). Quality criteria (RNA integrity) were assayed using an Agilent 2100 Bioanalyzer and RNA 6000 Nano kits (Agilent Technologies, California, USA). Neither the sampling time nor the surfacing time did affect RNA quality (Kruskal–Wallis rank sum test, Kruskal–Wallis chi-squared = 8.3992, df = 12, *p*-value = 0.7532; Wilcoxon rank sum test, W = 532.5, *p*-value = 0.1114). For both the in situ and the laboratory experiments, five animals were selected for each time point for further sequencing, making a total of 65 samples for each experiment. All electropherograms were individually checked, and sequenced samples had RIN comprised between 7.0 and 9.8, which meets high-quality sequencing standards.

**cDNA library construction and sequencing**. The study was conducted on gill tissues only, and specifically on the central part of the left gill. For RNA sequencing, the cDNA libraries were built for each sample using the TruSeq Stranded mRNA Sample Prep kit v2 (Illumina, California, USA) according to manufacturer instructions. To work on the transcriptome of the mussel only and get rid of bacterial RNA, mRNA were purified using poly-(T) beads from 3 μg of each total RNA sample, then cleaved in segments of 400 bp on average (350-450 bp range). Cleaved RNA fragments then were primed with random hexamers and reverse transcribed into first strand cDNA. A second strand of cDNA was consecutively synthesised, and double-stranded cDNA was purified using beads. The 3' ends of the blunt fragments then were adenylated. Indexed adapters were ligated to the cDNA fragments enriched by PCR (11 cycles). Libraries then were purified and quality-assessed using a Fragment Analyzer (Advanced Analytical Technologies, Inc., Iowa, USA). For each experiment separately, the 65 libraries were quantified by qPCR using the Kapa Library Quantification kit (Roche Sequencing, California, USA). They were then normalised, multiplexed on one S4 flow cell lane, and paired-end lengths of 2 × 150 bp were sequenced on an Illumina NovaSeq system (Illumina, California, USA) at the GeT-Plage core facility (Toulouse, France; http://get.genotoul.fr) producing an average number of 92.0 ± 29.3 and 93.2 ± 16.3 millions of read pairs per library for the in situ and laboratory experiments, respectively (mean ± sd, *n* = 2 × 65 for each experiment). For marine emerging model organisms including Molluscs, for which the genome is not available, at least 20

million reads are required for tissue samples intended for de novo transcriptome assembly[76]. Furthermore, transcriptome analysis of biological rhythms depends on read depths. There is no recommendation for marine species or ultradian (<20–28 h) rhythms, but 10–20 and 20–40 million reads have been recommended to detect 75–100% of circadian transcripts in flies and mice respectively[54].

**De novo transcriptome and quantification**. RNA-seq reads quality checking was performed manually by analyzing the fastQC (https://www.bioinformatics.babraham.ac.uk/projects/fastqc/ version 0.11.2) and contamination alignment results. Then, as there is currently no reference genome available for *B. azoricus*, reads have been assembled de novo to produce a reference transcriptome. Using too many reads in transcriptome assembly can lead to spurious contigs[77]; the following approach was therefore adopted: the assembly was tested with four small files from the in situ experiment. They were assembled individually with DRAP[78] (version 1.91) and then meta-assembled with the same software package to obtain a unique reference transcriptome. All the other read files, from both the in situ and the laboratory experiments, were then aligned to this reference. The BWA MEM (version 0.7.12-r1039) alignment rates were very good for all the files (alignment rates > 96% and properly paired rates >90%); therefore we did not perform another run of assemblies. The transcriptome assembly was also assessed using Benchmarking Universal Single-Copy Orthologs (BUSCO v3)[30].

The reference file was annotated. Open reading frames (ORFs) were extracted from the reference contig using TransDecoder[79] (version 3.0.0, standard parameters). The reference file was annotated with (1) interproscan (including GO; version 5), (2) diamond (version v0.9.9) on refseq_protein, and (3) KEGG.

Sample quantification was performed with the following steps. The reference was indexed with the Burrows-Wheeler Aligner (BWA index version 0.7.12-r1039, standard parameters) and reads were aligned with BWA-MEM (version 0.7.12-r1039, standard parameters). The alignment files then were compressed, sorted and indexed with SAMtools[80] view, sort and index (version 1.3.1, standard parameters). Mitochondrial and 18S and 28S ribosomal reads then were filtered out and all samples were randomly down-sampled[51] to the smallest library size (43.10⁶ paired-reads). Quantification was then realised with SAMtools idxstats (version 1.3.1, standard parameters) and result file composition was performed with unix cut and paste.

**Rhythm analyses**. Transcripts rhythmicity was investigated with an exploratory strategy: there was no standard approach to analyse these data, and authors often used previous knowledge and known profiles for specific rhythmic genes to determine which method fits with expected results[51]. We had no such knowledge for this work, we did not know whether deep-sea mussels still possess a biological clock and, if any, if the clock(s) is/are still functional in this peculiar environment. We therefore analysed data and crosschecked our results with different methods.

Contigs with 0 counts in at least 35 samples were filtered out of the analyses, which matched 34 and 56 contigs in the in situ and lab experiments, respectively. We used two normalisation methods: (1) we worked on the down-sampled data[54]; (2) we also normalised the data using DESeq2[39] (v 1.22.2) without using a zero-mean normal prior.

Rhythmicity analyses of the in situ and laboratory temporal transcriptomes were performed with R[81] (v 3.5.0), using RAIN[35] (v 1.16.0), and the ABSR[36] algorithms. RAIN was run using all replicates, while ABSR was run on the median value for each time point as the code is currently written for one time series. As the periodogram analyses revealed that 12.5 and 26.3 h were the dominant periods in the percentage of opened mussels, the transcriptome analysis was focused on tidal and daily rhythms. We considered both the tidal period 12.4 h and the circatidal range (10.4–14.4 h) for the analysis of tidal transcripts, and both the daily period 24.8 h (determined by the sampling interval) and the circadian range (20-28 h reduced here to 20–24.8 h because we sampled over 24.8 h only) for the analysis of daily transcripts. First, because one data point is one outbred animal, and we observed a high inter-individual variability in our datasets. Secondly, current algorithms do not provide a confidence interval for a specific period to be tested. Finally, in the darkness of the deep sea, a possible circadian clock at work in *B. azoricus* would be free-running while a putative circatidal clock would be free-running in the laboratory under L:D cycles without tides. RAIN and ABSR both detected rhythmic transcripts. The ABSR algorithm requires the setting of a threshold value for the spectral density, a crucial step that determines how conservative the method will be. Without prior knowledge about the species and its rhythmic behaviour, we selected the threshold by testing values between 0 and 30, and taking the mean inflection point for the number of tidal and daily rhythmic transcript curves. The threshold was calculated separately for the in situ and lab experiments, and actually set at 8 for both experiments and both normalisation methods (Supplementary Fig. 7). All *p*-values were adjusted for multiple-testing: the false discovery rate (FDR) was either included in the core code (ABSR), or it was done post-hoc with the Benjamini-Hochberg correction (RAIN). Transcripts were considered as rhythmic for FDR ≤ 0.05. Heatmaps were drawn with pheatmap (v 1.0.12) on median-normalised[54] transcripts expression. Global gene ontology analyses of rhythmic transcripts were performed using Blast2GO[82] (v.5.2.5); and gene enrichment analysis with the topGO package[83] (v.2.34.0).

**Identification of canonical circadian clock gene candidates.** Putative *B. azoricus* circadian clock gene sequences were deposited in Genbank. These sequences were taken from our reference transcriptome using nucleotidic and proteic BLAST (Basic Local Alignment Search Tool) with a selection of known circadian clock genes as queries (Supplementary Data 4a). While building the de novo transcriptome, about 3% of the reads were not included in the assembly because their expression was too low to pass the quality thresholds. A specific assembly, distinct from the reference transcriptome, was built and scanned as well as for the candidate circadian clock genes. In addition, non-published sequences were also analysed (A. Tanguy, co-author). Candidate sequences (Supplementary Data 4b–d) were confirmed manually by alignment with reference sequences and using Pfam conserved domains[84]. Deduced amino acid sequences were used for phylogenetic analysis. Isoforms were detected manually with MAFFT (multiple alignment fast Fourier transform) alignments[85] and only the longest isoforms were incorporated into phylogenetic analyses. Candidate circadian clock transcripts originating from the present sequencing data were included in the reference transcriptome for quantification; sequences provided by A. Tanguy were only used to increase the resolution of the phylogenetic analyses.

All trees were constructed with the same procedure. Sequences were aligned with MAFFT alignment algorithm (v7.388) using default parameters. Alignment cleaning was performed manually, keeping the sequences included between the first and last conserved domains of the considered clock gene for phylogenetic analyses (Supplementary Data 4e–h). Phylogenetic parameters (i.e., common substitution model) were set up using IQtree phylogenetic software[86] (v1.6.7), and performed with PhyML algorithms (v3.3.2), methods and utilities[87]. All trees then were constructed with the same parameters: Maximum-likelihood tree using LG substitution model, with tree topology, branch length and substitution rate optimisation, gamma estimation, and branch support values assessed using aLRT.

For homogeneity in phylogenetic analyses, the same taxonomic groups were represented: Lophotrochozoa, Insecta, Crustacea, and Euteleostomi (bony vertebrates). Outgroups were chosen to share as many conserved domains as possible and to be sufficiently distant to the gene considered. CPD I and II were chosen for the Cryptochrome/Photolyase tree because they present DNA photolyase and FAD binding seven domains[88]. Replication fork protection complex were chosen as outgroup for Timeless/Timeout tree because they share the timeless domain. Neuronal Pas Domain proteins were used as outgroup for the Clock/Bmal and Period trees because they share two bHLH and one PAS domain.

**Reporting summary.** Further information on research design is available in the Nature Research Reporting Summary linked to this article.

## Data availability

Metadata from the Momarsat 2017 cruise are available under the doi [doi.org/10.17600/17000500]. RNA-seq data were deposited in the ArrayExpress database at EMBL-EBI [www.ebi.ac.uk/arrayexpress] under accession number E-MTAB-8451. The ENA study accession is ERP117902. The data and de novo reference transcriptome are also in the SEXTANT database under the doi [https://doi.org/10.12770/971d2c1a-51cc-49fd-882c-465970de8ed2]. Circadian clock genes sequences were deposited in Genbank under the following accession: *bmal*, MN611450; *Clock*, MN597894; *period*, MN611455; *timeless*, MN611456; *timeout1*, MN611457; *timeout2*, MN611451; *cry1a*, MN611458; *cry1b*, MN611453; *cry2*, MN611452; *6-4 photolyase*, MN611454; *cry DASH*, MN611459. Source data are provided with this paper.

## Code availability

All software or packages used to analyse the data are mentioned in the "Methods" section. Source data are provided with this paper.

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

## Acknowledgements
The authors thank the commandant and crew of the "Pourquoi Pas?"; the team of the Momarsat 2017 cruise; the chief scientist of the Biobaz 2017 cruise Pr. François Lallier; the whole *Victor6000* team for sampling under red light on such a precise timing; Christophe Duchi; Philippe Rodier, Sandra Fuchs and Françoise Lesongeur for their help in the experimental setting and dissection; Dr. Ana Colaço for her help in storing and shipping the samples; António Godinho for his help for the Lab Horta experimental setting; Darryl Perrée for his help with molecular biology analyses; Dr. Mathilde Cannat for providing the field pressure data; Glyn Orpwood for correcting the English. The authors acknowledge the Pôle de Calcul et de Données Marines (PCDM) for providing DATARMOR storage and computational resources (ABSR and BLAST analyses). URL: http://www.ifremer.fr/pcdm. This work was supported by the Laboratoire d'Excellence LabexMER (ANR-10-LABX-19) and co-funded by a grant from the French government under the program Investissements d'Avenir, and by a grant from the Regional Council

of Brittany. This work was also supported by the Regional Council of Brittany for the project CHRONoS. The work was finally also supported by the project "Pourquoi pas les abysses?" of Ifremer. The project is part of the EMSO-Azores (http://www.emso-fr.org) regional node, and of the EMSO ERIC Research Infrastructure (http://emso.eu/).

## Author contributions

Project design and funding acquisition: A.M.M., M.M., J.S., P.M.S., A.H., and C.F. Experimental design: A.M.M., M.M., J.S., and P.M.S. Field experiment: A.M.M., M.M., J.S., and P.M.S. Lab Horta experiment: A.M.M. Deep-sea molecular biology protocol: A.M.M., M.M., J.S., C.D., and A.H. Behavioural analysis: M.M. RNA extraction and processing: A.M.M. cDNA library construction and sequencing: C.E. Bioinformatic analyses (de novo transcriptome and transcript abundance quantification): C.K. and A.M.M. Biostatistic and data analyses: A.M.M. Phylogenetic analyses of potential clock genes: V.A., A.M.M., G.V.M., and A.T. Data interpretation: A.M.M., M.M., J.S., A.H., C.F., C.K., and G.V.M. Manuscript writing, original draft: A.M.M. Manuscript review and editing: A.M.M., M.M., J.S., C.F., A.H., C.K., G.V.M., and P.M.S. All authors read, amended and approved the final manuscript.

## Competing interests

The authors declare no competing interests.
