## [Peer Review File · Nature Communications]

Reviewers' Comments:

Reviewer #1:

Remarks to the Author:

This is a fascinating manuscript exploring rhythms in gene expression in a deep sea mussel - and extraordinary technical feat. The manuscript requires some additional analyses, mainly comparing the authors' results to three key published papers on intertidal and subtidal mussels transcriptome cycles and this comparison may change some conclusions. These are:

<https://www.ncbi.nlm.nih.gov/pubmed/18848447>

<https://www.ncbi.nlm.nih.gov/pubmed/21911390>

<https://www.ncbi.nlm.nih.gov/pubmed/27538734>

If for example some of the rhythms described in the present report also occur in shallow mussels (which I believe the above papers show) the authors cannot conclude the results are due to deep ocean tidal flux, or other variables they map.

There are some other deep sea rhythm papers they should also read for a broader view, namely

Mercier A. & J.-F. Hamel 2014. Lunar periods in the annual reproductive cycles of marine invertebrates from cold subtidal and deep-sea environments. In: Numata H, Helm B (eds) Annual, lunar and tidal clocks: Patterns and mechanisms of Nature's enigmatic rhythms.

Mercier A., Z. Sun, S. Baillon & J.-F. Hamel 2011. Lunar rhythms in the deep sea: evidence from the reproductive periodicity of several marine invertebrates. *Journal of Biological Rhythms* 26: 82-86

Mercier A., Z. Sun & J.-F. Hamel 2011. Reproductive periodicity, spawning and development of the deep-sea scleractinian coral *Flabellum angulare*. *Marine Biology* 158: 371-380

Other than reading and considering these publications, I have two other issues that require addressing. One is more detail in the sampling methodology. Can the authors please explain how a sampled mussel was immersed in the stabilizing solution. Was seawater pumped out and stabilized pumped in? What was the time lag between sampling and surfacing? Were all samples collected and then surfaced together? Was time post sampling treated as a variable in data analyses? If one sample was sitting in solution for 24 hours longer than another this is a potential biasing factor. Of course the conditions are extraordinary- and achieving this sampling is an exceptional feat, but I'd like to hear details and steps taken to account for any such technical hurdles.

The other issue is data availability. The cited accession number for ArrayExpress does not find anything. I also searched for *Bathymodiolus* and once again did not find anything.

No mention is made of where the raw sequencing reads are deposited. I suggest the NCBI SRA for all primary read data.

Sampling

Can the authors please explain how a sampled mussel was immersed in the stabilizing solution. Was seawater pumped out and stabilizer pumped in? What was the time lapse between sampling and surfacing? Were all samples collected and then surfaced together? Was time post sampling treated as a variable in data analyses? How long does whole mussel RNA remain intact under these conditions?

Was a tissue biopsy collected? How?

Minor issues

Supp line 3. "Our work is the first ecological study of biological rhythms in a non-model organism in the deep-sea."

This isn't really correct- see papers above.

Reviewer #2:

Remarks to the Author:

Mat et al. analyze behavior and gill transcriptome of the deep-sea vent mussel *Bathymodiolus azoricus* from two different locations at the MAR (1688m and 834m depth). On-site behavior (in 2min sequences, every 6 hrs) and gill transcriptomes (every 2hr,4min) are recorded from mussels at 1688m, while *B.azoricus* from 834m were taken to the lab for transcriptomic analyses under a 12:12 LD regime. The author identify rhythmic transcripts with tidal and diel frequencies from the on-site samples, as well as the lab samples and also suggest that the on-site mussle behavior is rhythmic. Some core circadian clock genes cycle with periods of app. 10-16hrs. Based on these results the author conclude that biological rhythms exist in deep sea organisms- either driven exogenously or endogenously.

I discussed this manuscript with two advanced post-docs in the lab, one with a strong chronobiological background in conventional genetic lab systems and the other with a marine ecological background. We came to the following conclusions:

Showing clear transcriptional and behavioral rhythms for deep sea organisms would be a very important finding for marine ecology, even if otherwise functional data are lacking. But there are several major concerns about the manuscript:

- 1.) We understand that the authors managed to develop a protocol that fixes the specimen directly at the deep sea sampling side. Is this correct- then please make this very clear in the text, because this would be a major break-through. Please also provide more details and the full protocol for this. If not- then this strongly de-values the findings described in the manuscript, because a lot of changes occur while getting the samples up to the boat and this could certainly cause rhythms to be detected artificially (e.g. by passing through the photic layer). Thus, this points needs to have an absolutely clear and detailed description.
- 2.) Given the value of the samples and the sentence in the methods that the different tissues were all stored in separate tubes and processed, we do not believe that gills were the only tissue sequenced from the on-site sampling. What happened to the other tissues? Report any other transcriptomes and their potential changes/rhythms that were observed. If rhythms were only visible in the gills that would not necessarily devalue the authors' findings, but must be reported and discussed!
- 3.) How about the data from the symbionts? Any rhythms detected there? Report and discuss, please. No matter what the outcome- this is very valuable and important information to understand what is going on at the deep sea.
- 4.) How specific are the about tidal and diel rhythms in the deep sea samples? In other words, if the authors had looked for rhythms with 18-20hr periods or 17-19hr periods- would they also be present? Which other period lengths exist and which percentage of the transcripts are exhibiting those?

5.) Performing a lab experiment under LD 12:12 when looking for 12 or 24hr rhythms is pretty pointless, as the artificial light cycle might now well cause the transcript changes. A DD experiment would have been much more appropriate or- if the authors believe that the mussels would already likely be de-synchronized in the lab- mimic the pressure changes or small temperature fluctuation for potential entrainment/ rhythm generation and compare to data obtained from the field sampling. This is all technically feasible. The LD 12:12 is just biologically totally artificial and hence at present useless.

DD would also allow for observing the mussel behavior in the lab and look for rhythmic components.

6.) The title and introduction are full of overstatements- tune them down. There are multiple papers already in the literature that have started to look at rhythms in the deep sea. (just to mention two examples ignored by the authors:

- Deep Sea Research Part I: Oceanographic Research Papers Vol 54 (11), Nov 2007, pp 1944-1956

<https://doi.org/10.1016/j.dsr.2007.08.005>;

-PLoS One. 2017 May 26;12(5):e0178417. doi: 10.1371/journal.pone.0178417.)

"the deep sea is our planet's largest biome"- not sure this superlative is correct. How about all the gut areas of all the human on the planet? And how to really measure and compare such statements?

Minor, but still important, points:

1.) Fig.2c- Y-axis labels missing

2.) Fig.2: b vs. f and e vs. i- which transcripts are significantly rhythmic under both conditions? (or are these in majority different?)

3.) Fig.3a "organelle organization" occurs twice- doesn't make sense

4.) EDF 1: move to main Figures,

EDF1b- X-axis label is not very intuitive. The sampling was done only for 2mins every 6 hours- this is not at all apparent from the current X-axis. Please display more correctly. Alternatively- plot data for each day separately.

How can the authors conclude that there are rhythms in the data? What is the period length of those?

5.) Depending on scientific field the label "in situ" describes something totally different. Please use other words, e.g. "field samples" to label these experiments.

6.) EDF 7: This EDF is poorly described in the text, albeit pretty important. Move those graphs that show significant rhythms from the field samples to the main figures and describe and depict them well! Furthermore, in EDF6 the authors show that *B. azoricus* has two cry1 genes (and proteins). What are the transcript changes for the second cry1 gene?

7.) There are two erratic "?" in the acknowledgement section.

Reviewer #3:

Remarks to the Author:

This is a very interesting manuscript, that expands our knowledge of the biological rhythms to the deep-sea hydrothermal vent invertebrates. In order to support the conclusion, the authors have

successfully collected very high-quality behavioural and gene expression data, both of which show biological rhythms with statistical power. In addition, the usage of red light during the *Bathymodiolus* sample collection as well as the deep-sea in situ RNAlater fixations is highly welcome in deep-sea biology. The RNA-Seq sample preparation and also the downstream statistics are state-of-the-art and stringent. I would thus consider the conclusion very convincing and exciting.

My concern with the experimental part is the laboratory experiment. I understand that the authors would like to find out whether the response to light is an evolutionary relic. However, please note that the deep-sea *Bathymodiolus* mussel has diverged from shallow-water mussel at least 100 million years ago (Lorion et al. 2013; Sun et al. 2017). Light is a non-existing stress/factor in deep-sea *Bathymodiolus* mussel lifetime, and the biological response to light is therefore unreal. The including of this experiment is a bit distracting, and I would suggest the authors remove this part because the rest result of behaviour and in situ RNA expression data are collectively very convincing already.

In addition, although the overall sampling and experiments are in general well-performed, the manuscript is not very well-written particularly the "Discussion" part, possibly due to the limited words in the current letter format. Notwithstanding, it should be remarked that it is full of thought-provoking observations. In my opinion, from the evolutionary perspective, it would be interesting to compare the result from this study with similar biological rhythms research on shallow-water marine invertebrates, at least with mussels (e.g. Connor & Gracey, 2011), to shed light on the whether the deep-sea *Bathymodiolus* mussel are using similar or dramatically different molecular mechanism in order to find out whether there is a universal tool-kit of biological rhythms.

Minor comments:

One textual issue is the use of 'deep sea' and 'deep-sea'. If you are speaking about a habitat like the deep sea, do not use a hyphen between deep and sea. Please check your complete manuscript. Therefore, remove the hyphen in the Title and also on page 2 line 19, 30, 342, 344, 393 and Figure 1 legend etc.

Line 410, 700 g? or 700 mg? Plus, when the mussels were fixed in situ, did the authors try to break the shell to make sure the RNAlater could fully penetrate and fix the tissue?

Reviewers' comments:

Reviewer #1 (Remarks to the Author):

This is a fascinating manuscript exploring rhythms in gene expression in a deep sea mussel - and extraordinary technical feat. The manuscript requires some additional analyses, mainly comparing the authors' results to three key published papers on intertidal and subtidal mussels transcriptome cycles and this comparison may change some conclusions. These are:

<https://www.ncbi.nlm.nih.gov/pubmed/18848447>

<https://www.ncbi.nlm.nih.gov/pubmed/21911390>

<https://www.ncbi.nlm.nih.gov/pubmed/27538734>

If for example some of the rhythms described in the present report also occur in shallow mussels (which I believe the above papers show) the authors cannot conclude the results are due to deep ocean tidal flux, or other variables they map.

>> We added a full paragraph in the Discussion to compare our results with those obtained in shallow-water bivalves, mainly mussels (L318-331). This involves mentioning the work from Connor and Gracey in *Mytilus californianus*, plus other research.

We actually do not conclude that the oscillations we observed in *B. azoricus* are due to deep ocean tidal flux. We have modified a sentence that may have led to confusion L154-155. The periodicity observed in mussels in the field can either 1) represent a direct response to an environmental signal without involving any internal clock, or 2) be a "true" biological rhythm driven by an internal clock synchronised by environmental signals. In both cases, it implies that mussels can perceive a tidal signal in the deep sea (current, or temperature, or food/chemical compounds availability, etc). Therefore, even if we cannot state whether the rhythms reported in *B. azoricus* have an exogenous or endogenous origin, the fact that the hydrothermal vent is under tidal influence is crucial information. This is now clearly explained in L300-309. This information is also important to shed light on the functional significance of the temporal organisation of *B. azoricus* in its ecosystem.

The comparison with shallow-water bivalves is certainly relevant but however does not change our conclusions. Tidal and daily rhythms have been reported in both deep-sea and shallow-water bivalves. Both types of environments are submitted to tidal influence. The mechanisms underlying rhythmicity in marine species in general, and in mussels in particular, remain unclear. More than 20 years ago, Naylor (1996, Q Rev Biol 71: 586) mentioned that "A key question is whether the biological basis of tidally rhythmic behaviour and physiology can be accommodated with diel rhythmicity in a common conceptual framework". That question still remains open. However, deep-sea ecosystems, totally blind to the Light:Dark cycle but exposed to tidal stimuli, could therefore provide a unique opportunity for untangling biological clocks in marine species (L332).

There are some other deep sea rhythm papers they should also read for a broader view, namely

Mercier A. & J.-F. Hamel 2014. Lunar periods in the annual reproductive cycles of marine invertebrates from cold subtidal and deep-sea environments. In: Numata H, Helm B (eds) Annual, lunar and tidal clocks: Patterns and mechanisms of Nature's enigmatic rhythms.

Mercier A., Z. Sun, S. Baillon & J.-F. Hamel 2011. Lunar rhythms in the deep sea: evidence from the reproductive periodicity of several marine invertebrates. Journal of Biological Rhythms 26: 82-86

Mercier A., Z. Sun & J.-F. Hamel 2011. Reproductive periodicity, spawning and development of the deep-sea scleractinian coral *Flabellum angulare*. Marine Biology 158: 371-380

>> Reviewer 1 is right, other papers have provided great information on the possibility of biological rhythms in deep-sea organisms, and we now mention several of them and open the Introduction to a broader perspective (L65-72). Specifically, we have also taken into account the above mentioned publications.

Other than reading and considering these publications, I have two other issues that require addressing. One is more detail in the sampling methodology. Can the authors please explain how a sampled mussel was immersed in the stabilizing solution. Was seawater pumped out and stabilized pumped in? What was the time lag between sampling and surfacing? Were all samples collected and then surfaced together? Was time post sampling treated as a variable in data analyses? If one sample was sitting in solution for 24 hours longer than another this is a potential biasing factor. Of course the conditions are extraordinary- and achieving this sampling is an exceptional feat, but I'd like to hear details and steps taken to account for any such technical hurdles.

>> As requested by all 3 reviewers, we now provide additional details on the sampling procedures (L 106, L387-392, L395-405, and 425-431) plus video sequences recorded by the ROV *Victor6000* during the *in situ* sampling (Supp Movies 1 and 2) to be absolutely clear about the sampling protocol. We think that these videos are very informative and clearly answer the Reviewers' comments and would be of great interest to the readers.

To specifically address Reviewer 1s' questions:

- a separate sampling box was used for each sampling time. Each sampling box had a useful volume of ~5 L and had been filled to the brim on board with a non-toxic in-house RNA stabilising solution. The stabilising solution is kept in the box when it is opened for collecting the samples on the hydrothermal edifice because: 1) the density of the solution (~1.3 kg/L) is higher than the density of seawater (~1.02 kg/L); and 2) the boxes were protected, on the inside, by rubber "blades" which are specific seals that hold the solution and the samples inside the box (L390-392, plus Supp. Movie 2).

- For each sampling time, 7 to 15 mussels were collected and placed in a box to preserve RNA immediately upon sampling. As the volume of one mussel is about 50 ml, tissues were thus placed in 5-10 volumes of RNA stabilising solution, which matches requirement for RNA preservation (US patent 8178296 B2). Because the shells were intentionally slightly cracked open using the claw of the ROV arm and then they were placed in the box, the RNA stabilising solution rapidly permeated mussel tissues to stabilise and protect cellular RNA, avoiding the need to immediately process tissue samples (L399-401). This was confirmed by the high quality of RNA extracts we obtained (L425-431).

- The samples were brought to the surface in two stages: the 3 first sampling times (0 h, 2 h 04 min, and 4 h 08min) were brought up together 5 h 36 min after the beginning of the deep-sea sampling while the remaining samples were surfaced together at the end of the sampling, 27 h 23 min after the beginning of the experiment. Time lag between sampling and surfacing varied from 5 h 36 min to 21 h 13 min (L402-404).

This two-step surfacing was due to technical constraints. It takes 45 minutes for the ROV to dive down to 1,700 m and 45 minutes to resurface, and once the ROV surfaces, it goes into maintenance mode for about 8 hours, making it impossible to sample and surface every two hours. We therefore used an elevator to surface part of the samples, but it cannot be surfaced at any time during the dive. These constraints, related to the carrying and operational capacities of both the elevator and the ROV, have to be taken into account. Also, the depressurisation and sea surface pressure are at best stressful and more often lethal for deep-sea specimens (Shillito et al., 2014, *J Mar Sci Technol* 22:97-102). To work around these constraints, well-known by deep-sea scientists, we developed the approach to fix samples directly on the seafloor. Once in the preservation solution, the RNAs no longer change and remain thus preserved before dissection.

Moreover, the time post sampling may be considered as negligible because:

1. the in-house RNA stabilising solution is a fixative solution that allows RNA isolation from intact tissues;
2. the ROV pilots used the mechanical arm to break mussel shells upon collection to allow rapid infiltration of the solution into each individual. Once on board, mussels with unbroken shells were discarded (added now L401);
3. samples can be stored in this solution at 25°C for up to one week, at 4°C for up to one month, and indefinitely at -20 or -80°C without RNA degradation. Our samples were kept both in the deep sea and on board at 4°C before dissection, and at -20°C or -80°C after dissection, matching thus the storage requirements as performed for all molecular samples (outdoor experiments, coastal sampling);
4. the gene enrichment analyses that we ran on the rhythmic transcripts detected for the *in situ* experiment did not show gene ontology terms related to "apoptosis" or "cell death" (expected in the case of inappropriate storage);
5. finally, we carefully controlled for RNA quality and integrity using a Nanodrop and a Bioanalyzer as stated in the text (L422-431). RNA is extremely sensitive to degradation. We are highly experienced with RNA extraction in the lab, any break in the cold chain or any lasting dissection without RNA stabilising solution or cold temperatures results in degraded RNA. All samples exhibited OD260nm/OD280nm and OD260nm/OD230nm > 1.8 (OD: optical density). These ratios indicate the purity of the samples. RIN (RNA Integrity Number) is a metric commonly used to assess RNA quality. RNA yield and quality were homogeneous between the different sampling points: neither the sampling time nor the surfacing time did affect the RIN (Kruskal-Wallis rank sum test, Kruskal-Wallis chi-squared = 8.3992, df = 12, *p*-value = 0.7532; Wilcoxon rank sum test, *W* = 532.5, *p*-value = 0.1114). All the sequenced samples had RIN comprised in the interval 7.0-9.8, indicating high quality RNA. This has been added L425-431.

The other issue is data availability. The cited accession number for ArrayExpress does not find anything. I also searched for *Bathymodiolus* and once again did not find anything.

No mention is made of where the raw sequencing reads are deposited. I suggest the NCBI SRA for all primary read data.

>> Indeed, the RNA-seq data have been deposited at EMBL-EBI in the ArrayExpress database under accession number E-MTAB-8451, as mentioned in the Data availability section (L546). The ArrayExpress database is the EMBL-EBI archive for high-throughput functional genomics experiments, and this archive is part of the EMBL-EBI European Nucleotide Archive (ENA). We added the ENA study accession number in the manuscript (L548). Reviewer 1 may not have been able to find the data because its access is currently restricted and the data will be released upon article publication. However, we fully understand the will for Reviewer 1 to confirm data availability. Please find below the link and codes for accessing the data on the ArrayExpress website:

<http://www.ebi.ac.uk/arrayexpress/experiments/E-MTAB-8451>

using the following login details:

Username: Reviewer_E-MTAB-8451

Password: Uagswh5q

Nature journals suitable repositories for DNA and RNA sequences are EMBL Nucleotide Sequence Database (ENA/EBI, Europe), Genbank (USA), and DDBJ (DNA DataBank of Japan). All three collections are coordinated by the International Nucleotide Sequence Database Collaboration (INSDC). They share their data daily. Upon release, our raw sequences will thus be available on the three databases.

The transcriptome file and the raw reads are also available on SEXTANT (L549), a database for marine data. DOI: 10.12770/971d2c1a-51cc-49fd-882c-465970de8ed2

Finally, the circadian clock genes sequences have been deposited in Genbank. Here again, access will not be available prior to publication and data release. There is no code to access these data before they are released, so please find at the end of this document the emails that confirm the sequences submission received by Genbank after we submitted our sequences.

To avoid any confusion, we modified the phylogenetic trees to use the accession numbers provided for our sequences immediately in the results, and not only in the Data availability section.

For both the data submitted to ArrayExpress and Genbank: it is mandatory to set a date until which data are kept confidential. Data will not be released to the public database until this date, or until the data or accession numbers appear in print, whichever is first. So, if the article is accepted before the date, the data will be released upon publication.

Sampling

Can the authors please explain how a sampled mussel was immersed in the stabilizing solution. Was seawater pumped out and stabilizer pumped in? What was the time lapse between sampling and surfacing? Were all samples collected and then surfaced together? Was time post sampling treated as a variable in data analyses? How long does whole mussel RNA remain intact under these conditions? Was a tissue biopsy collected? How?

>> We address these questions above in details. As mentioned, additional details on sampling procedures and video sequences have been added to the manuscript.

Minor issues

Supp line 3. "Our work is the first ecological study of biological rhythms in a non-model organism in the deep-sea."

This isn't really correct- see papers above.

>> We agree with Reviewer 1, this was not specific enough. What we meant and clarify now L279-282 is that " Our work is the first deep-sea molecular study of biological rhythms in a non-model organism performed while directly fixing specimens *in situ*. We have developed a cutting-edge protocol to work both under realistic ecological conditions and follow at best latest guidelines for "omics" analyses in chronobiology."

Reviewer #2 (Remarks to the Author):

Mat et al. analyze behavior and gill transcriptome of the deep-sea vent mussel *Bathymodiolus azoricus* from two different locations at the MAR (1688m and 834m depth). On-site behavior (in 2min sequences, every 6 hrs) and gill transcriptomes (every 2hr,4min) are recorded from mussels at 1688m, while *B.azoricus* from 834m were taken to the lab for transcriptomic analyses under a 12:12 LD regime. The author identify rhythmic transcripts with tidal and diel frequencies from the on-site samples, as well as the lab samples and also suggest that the on-site mussle behavior is rhythmic. Some core circadian clock genes cycle with periods of app. 10-16hrs. Based on these results the author conclude that biological rhythms exist in deep sea organisms- either driven exogenously or endogenously.

I discussed this manuscript with two advanced post-docs in the lab, one with a strong chronobiological background in conventional genetic lab systems and the other with a marine ecological background. We came to the following conclusions:

Showing clear transcriptional and behavioral rhythms for deep sea organisms would be a very important finding for marine ecology, even if otherwise functional data are lacking. But there are several major concerns about the manuscript:

1.) We understand that the authors managed to develop a protocol that fixes the specimen directly at the deep sea sampling side. Is this correct- then please make this very clear in the text, because this would be a major break-through. Please also provide more details and the full protocol for this. If not- then this strongly de-values the findings described in the manuscript, because a lot of changes occur while getting the samples up to the boat and this could certainly cause rhythms to be detected artificially (e.g. by passing through the photic layer). Thus, this points needs to have an absolutely clear and detailed description.

>> We indeed developed a protocol to fix specimens directly at the deep-sea sampling site under red light, allowing a realistic approach on the molecular processes that occur. We did so because we agree with Reviewer 2: a lot of changes occur while bringing live samples back on board. For example, it takes at least 1h to bring samples back on board from 1,700 m depth. Also, the depressurisation and sea surface pressure are at best stressful and more often lethal for deep-sea specimens (Shillito et al., 2014, J Mar Sci Technol 22:97-102). Therefore, fixing samples on board would ruin any molecular rhythm analysis: the ecologically relevant physiology signals of the mussel would likely be overwhelmed by stress and lethal signals, and it would likely be difficult to detect any rhythmic signal. Using red light was also key to observe a realistic ecological signal, as evidenced by the laboratory experiment (details below in Reviewer 2 major comment 5).

As requested by all 3 reviewers, we now provide additional details on the sampling procedures (L 106, L387-392, L395-405, and 425-431) plus videos recorded during the sampling *in situ* (Supp Movies 1 and 2) to be absolutely clear and informative about the sampling protocol. Please see also above on page 2, we provide in-depth clarification to this question in the answers to Reviewer 1.

2.) Given the value of the samples and the sentence in the methods that the different tissues were all stored in separate tubes and processed, we do not believe that gills were the only tissue sequenced from the on-site sampling. What happened to the other tissues? Report any other transcriptomes and their potential changes/rhythms that were observed. If rhythms were only visible in the gills that would not necessarily devalue the authors' findings, but must be reported and discussed!

>> Actually, this is the truth: gills tissues were the only tissues sequenced, both from the *in situ* and lab sampling experiments. While we indeed dissected the foot and conserved them in commercial RNeasy® RNA later, and stored the rest of the mussels in-house RNA stabilising solution, we did not process them. For the gills, we carefully dissected the very same part of the tissue, *i.e.* the central part of the left gill. We selected the gills because:

1. gills are a key organ for both nutrition and respiration in marine bivalves, and are the first sensor of the environment as stated L107-108;
2. Information on rhythms in mussels had already been obtained from littoral species, and when a specific tissue was targeted, it has always been the gills (e.g. Connor and Gracey, 2011, 2012; these authors are now mentioned in the Discussion L322-323).

Working on the gills thus maximised the chances of success of our research and allows comparisons with previous studies in coastal species. This is now more clearly explained in the revised version of the manuscript (L406 and L433).

We totally agree with Reviewer 2: if rhythms were only visible in the gills, that would not devalue the results. It has been shown for circadian rhythms in terrestrial species that rhythms in gene expression are tissue-dependent (L282-284). Here, we obtain conclusive results of rhythmicity in the gills, but this work is not exhaustive, as is the case in most of the studies on rhythms in marine coastal species and even terrestrial species. As stated L288, our data clearly show that biological rhythms influence mussels' physiology. However, capturing the full impact of rhythms on the transcriptome is another challenge that would require analysing every tissue over a longer time period, and consequently, additional (wo)manpower and funds. It can lead to the production of an atlas of rhythms as performed in mice (Zhang et al, 2014, PNAS, 111: 16219–16224).

3.) How about the data from the symbionts? Any rhythms detected there? Report and discuss, please. No matter what the outcome- this is very valuable and important information to understand what is going on at the deep sea.

>> We actually did not perform the symbionts transcriptome. Bacterial RNA was discarded during library preparation by selecting mRNA on poly-T beads (after RNA quality check); this was confirmed by quality controls. This was mentioned in the Method section L435-436 and, the Results section L117-119 and L171-175. However, we understand this was not clear enough in the manuscript and we now clarify this L435: "To work on the transcriptome of the mussel only and get rid of bacterial RNA, mRNA were purified using poly-(T) beads...". Additionally, without specific steps to isolate bacterial RNA, the RNA extraction protocol we used leaves extremely small amounts of informative bacterial RNA (Westermann et al, 2017, PLoS Pathog 13(2): e1006033). We briefly discuss this aspect in the manuscript to open a perspective for future work (L173 and L265).

Indeed, working on the mussel transcriptome was already enough of a challenge as a first exploratory research project. Showing that rhythms exist in deep-sea mussels now raises exciting questions that we would like to investigate in the coming years. One of these questions relates to the relationship between the mussel and the bacteria in the rhythmic functioning of the holobiont as indicated L173. This is now added in the perspectives of this work (L265-267). It will require extra time and protocol developments for triple RNA extraction and bioinformatic analyses, but this information will be important in understanding *B. azoricus* and symbiotic interaction on hydrothermal vents.

4.) How specific are the about tidal and diel rhythms in the deep sea samples? In other words, if the authors had looked for rhythms with 18-20hr periods or 17-19hr periods- would they also be present? Which other period lengths exist and which percentage of the transcripts are exhibiting those?

>> As we were interested in tidal and daily rhythms based on our initial hypotheses, we specifically sampled and looked for rhythms within the circatidal (10.4-14.4h) and circadian (20-28h reduced here to 20-24.8h because we sampled over 24.8h only due to time constraint on the seafloor) ranges. This is detailed L495-498. It is a common practice to look for specific intervals rather than search for all potential periods in a dataset. We had several reasons to do so:

1. the Dutilleul multi-frequential periodogram analysis revealed that 12.5h and 26.3h were the dominant periods in the percentage of opened mussels (L96-98);
2. the RAIN method can only test for periods that are multiples of the sampling interval. So, we could only test for 18.6h within the 17-20h interval. The ABSR method does not have this limitation though;
3. testing more periods or intervals reduces the statistical power because we have to do extra corrections for multiple testing;
4. ultradian (<20h) and infradian (>28h) rhythms have already been reported in oysters for example (e.g Mat et al, 2012, Chronobiol Int 29(7): 857-867; Payton et al, 2017, Sci Rep 7: 3480), with periods ranging from 4 to >72h. While ultradian 10.4-14.4h rhythms in marine organisms can be generated by an endogenous circatidal clock (Zhang et al, 2013, Curr Biol 23: 1863-1873), ultradian rhythms have also and mainly been shown in terrestrial species. Specifically, 6, 8 or 12h- period rhythms are usually considered as harmonic of circadian rhythms (Castellana et al, 2018, Front Physiol 9:1178). Despite the large description of ultradian cycles and the recognition of their functional importance (van der Veen and Gerkema, 2017, FASEB J 31(2):743-750), the origin and mechanisms of these ultradian rhythms are globally unknown. For periods out of the circatidal or circadian ranges or their harmonics, or that do not correspond to any known physical cycle in the environment, it would be difficult to understand their meaning in the physiology of the animal at this stage.

So, given the loss of statistical power that these testings would generate, the fact that our working hypotheses relied on potential circatidal and circadian rhythms in *B. azoricus*, the fact that this is the first high-resolution molecular study on a vent species, and the fact that we could not derive much information from rhythms of other period ranges that do not correspond to any known physical cycle in the environment, we focused on the circatidal and circadian ranges. This is now mentioned in L493-495.

5.) Performing a lab experiment under LD 12:12 when looking for 12 or 24hr rhythms is pretty pointless, as the artificial light cycle might now well cause the transcript changes. A DD experiment would have been much more appropriate or- if the authors believe that the mussels would already likely be de-synchronized in the lab- mimic the pressure changes or small temperature fluctuation for

potential entrainment/ rhythm generation and compare to data obtained from the field sampling. This is all technically feasible. The LD 12:12 is just biologically totally artificial and hence at present useless. DD would also allow for observing the mussel behavior in the lab and look for rhythmic components.

>> We really believe that a L:D 12:12 lab experiment is not pointless at all and provide a real scientific value to the current study because:

1. working under red light *in situ* could be seen as a safe precaution, but without the L:D experiment, we could still question its validity for animals that are not exposed to light anymore. Working under red light with a ROV was a real challenge: visibility was reduced to a maximum of 4m, compared to ~15m under regular white light (see also Supp. Movie 1). The sampling was not trivial under these conditions: given the value of a ROV, its safety cannot be compromised. So, the importance of a biological safety precaution could be questioned compared to the technical risk it represents. The lab experiment showed that *B. azoricus* physiology is indeed different under L:D 12:12 and *in situ*: daily oscillations became more prominent than tidal oscillations. This supports and validates the need to work under red light for the deep-sea experiment, which is a fundamental result to share.
2. this suggests that *B. azoricus* can perceive light. We agree with Reviewers 2 and 3, this response is artificial or unreal as light is usually not a stress factor for deep-sea mussels. We do not want to claim that the lab experiment was ecologically realistic, and we state this very clearly now in L204 and L274-276. However, it has implications for future research, sampling, and management strategies. While current deep-sea research and investigations are performed under white light for technical reasons, this introduces a potentially great bias in any biological observation. We clarified this in L276-278.
3. our work was exploratory, and the first investigation of the temporal transcriptome in a non-model organism directly performed *in situ*, in the deep-sea. As stated in the text (originally in the Supp. Discussion, now mentioned in the article L140), in the absence of internal control, *i.e* a rhythmic gene whose expression is cyclic and whose profile is known, we took extra precautions to validate the data because everything is to be explored about biological rhythms in this species. The fact that we observed different patterns very consistent with the environmental signals under both *in situ* and laboratory conditions—tidal cycles predominated in the transcriptome and physiology of mussels sampled directly at hydrothermal vents under tidal influence, whereas daily cycles prevailed in mussels sampled in the lab under L:D cycles—contributes to making data collectively very convincing.
4. both Reviewers 1 and 3 asked for comparison with research published on shallow-water mussels. With all the precautions needed, because bringing deep-sea mussels back to the surface and housing them in the laboratory is a major change, the L:D experiment is important because it provides key insights to putting our work into perspective with littoral species submitted to both tides and L:D cycles (see now L318-326). The main reason is that the circatidal and circadian clocks could be described by one system, or at least share common regulators, and that question is still open in general and in mussels in particular (see now L326-331). It thus appears essential to study both the circatidal and circadian clockwork(s) in parallel. Here, while the inter-individual variability between mussels is too high to see a clear pattern, we still report that several potential circadian clock genes oscillate with a ~semi-diurnal period *in situ*, and the gene *period* also showed a circatidal period in the lab (Figure 5). We also observed transcripts with a circadian rhythmicity *in situ*, without known daily cycle, and transcripts with a circatidal rhythmicity in the lab without tidal signal. It suggests that both tidal and daily rhythms could be endogenously generated in *B. azoricus*. Is there indeed one or two clocks? That is now one perspective of this research.

We nevertheless appreciate that this experiment can be considered of less importance and groundbreaking science for the reader, as stated by Reviewer 3. But as explained above, it brings significant inputs to the whole research and to the deep-sea experiment. Therefore, we propose to compromise by moving most of the figures related to the L:D experiment to the Extended Data, thus emphasising the *in situ* work. There is one exception with Figure 5, as its data are important for the comparison with shallow-water mussels. We organised the text and main Figures accordingly.

We agree that both a D:D (Dark:Dark, constant darkness) experiment and an experiment under tidal entrainment would provide great information as well. However, these experiments remain challenging

and were not performed in this pilot experiment to explore the question of biological rhythms in *B. azoricus* mainly due to two points:

1. while this is indeed technically feasible, it is not trivial. Pressure and temperature data recorded *in situ* indicate that the Lucky Strike hydrothermal vents are under tidal influence. However, for an experiment under tidal entrainment, it first requires to identify which environmental signal(s) *B. azoricus* can perceive. Indeed, this is species-dependent and different environmental signals have been reported to be tidal synchronisers: water current (e.g. Mat et al, 2014, Mar Biol 161: 89-99), salinity (e.g. Reid and Naylor, 1990, J Biol Rhythms 5:333-347), temperature (e.g. Reid and Naylor, 1990), hydrostatic pressure (e.g. Reid and Naylor, 1990), water depth cycles (Chabot et al., 2008, Biol Bull 215:34-45), cycles of immersion and exposure to air (Williams and Naylor, 1969, J Exp Biol 51:715-725), vibration (Zhang et al., 2013, Curr Biol 23: 1863-1873), and turbulence cycles simulating wave action (Klapow, 1972, J Comp Physiol 79:233-258) and food pulses (Williams and Pilditch, 1997, J Biol Rhythms 12:173-181). In addition, at vents, tides generate reversal in bottom currents that modify the influence of the vent fluid on organisms. Depending on the tidal phase, organisms will be under varying hydrothermal vent fluid influence, so additional signal specific to vents could be sulphide or metal concentrations known to have a strong influence on species physiology at vents (e.g. Luther et al., 2001, Nature 410: 813-816; Schmidt et al., 2008, J Shellfish Res 27(1): 79-90). These variables are clearly difficult to control in the lab.

Additionally, if rhythms in marine organisms are generally robust under entrainment, they have often been reported as labile under free-running conditions, *i.e.* constant darkness or constant light (e.g. Last et al., 2009, Chronobiol Int 26:167-183; Mat et al, 2012, Chronobiol Int 29(7): 857-867; Palmer, 1995, The biological rhythms and clocks of intertidal animals, Oxford University Press). There is also a risk, as indicated by Reviewer 2, that mussels would be de-synchronized upon arrival in the lab. So this could require re-entraining the mussels first to synchronise them before releasing them into constant conditions.

For these types of experiments, especially those in constant conditions, it is usually ideal to monitor a non-invasive variable to ensure that animals are rhythmic before sacrificing animals. One of these parameters is behaviour for example (Mat et al, 2012, 2013, 2017). However, it first requires several days of acclimation to stabilise animals' behaviour after sampling, then several days, ideally a week, of data to have robust statistics, especially in constant conditions. These ideal conditions mean that these experiments take time to be performed. Combined with the fact that housing deep-sea mussels in the lab is technically and physiologically feasible, but that animals cannot be collected as easily as littoral species, these experiments remain extremely challenging for deep-sea organisms and were not performed in this pilot experiment to explore the question of biological rhythms in *B. azoricus*.

2. These experiments appear particularly interesting now with the current results. Now that we know that *B. azoricus* exhibits tidal and daily frequencies in valve behaviour and gill transcription both *in situ* and in the lab, the question is: does this represent true biological rhythm(s) driven by (an) internal clock(s), or does it simply reflect a direct response to environmental signals (L26-29, L300-303)? This is one of the perspectives of the current work.

6.) The title and introduction are full of overstatements- tune them down. There are multiple papers already in the literature that have started to look at rhythms in the deep sea. (just to mention two examples ignored by the authors:

– Deep Sea Research Part I: Oceanographic Research Papers Vol 54 (11), Nov 2007, pp 1944-1956 <https://doi.org/10.1016/j.dsr.2007.08.005>;

–PLoS One. 2017 May 26;12(5):e0178417. doi: 10.1371/journal.pone.0178417.)

>> The key here for us was "in the deep sea", compared to "in deep-sea specimens", as previously reported. We agree that it was not stated specifically enough and this has been corrected L279 for example. We revised potential overstatements in the whole manuscript, including in the abstract (L20 and L28). We agree that other papers have provided great information on the possibility of biological rhythms in deep-sea organisms, and we now mention several of them and open the Introduction to a broader perspective (L65-72).

We also now propose the following title: "Time is ticking in the deep sea: biological rhythms in the hydrothermal mussel *Bathymodiolus azoricus*."

"the deep sea is our planet's largest biome"- not sure this superlative is correct. How about all the gut areas of all the human on the planet? And how to really measure and compare such statements?

>> We used the word biome here as an ecological and biogeographical concept (cf Mucina, 2019,

New Phytol 222:97–114). But we understand that scientists in different fields can have a broader view of the word biome. And we agree that it would be difficult to measure and compare with for example all the gut of all human beings. We changed the sentence to " one of our planet's largest biomes " (L20).

Minor, but still important, points:

1.) Fig.2c- Y-axis labels missing

>> This has been corrected. Figure 2 is now Figure 3 because EDF 1 has been moved to main Figures (see below).

2.) Fig.2: b vs. f and e vs. i- which transcripts are significantly rhythmic under both conditions? (or are these in majority different?)

>> For the transcripts that were rhythmic in the range 10.4-14.4 h, 152 transcripts were rhythmic under both in situ (2,502 rhythmic transcripts) and laboratory conditions (1,988 rhythmic transcripts). For the transcripts that were rhythmic in the range 20-24.8 h, 63 transcripts were rhythmic under both in situ (869 rhythmic transcripts) and laboratory conditions (2,511 rhythmic transcripts). This information has been added in the text L199-203 and in a new Supp. Table 3.

3.) Fig.3a "organelle organization" occurs twice- doesn't make sense

>> We thank Reviewer 2 for underlining this non-sense. This has been corrected in Fig.3a which is now Fig.3F: the first occurrence is actually "Cellular protein modification process" (9%) instead of "Organelle organisation" that occurs later, at 6%.

4.) EDF 1: move to main Figures,

EDF1b- X-axis label is not very intuitive. The sampling was done only for 2mins every 6 hours- this is not at all apparent from the current X-axis. Please display more correctly. Alternatively- plot data for each day separately.

How can the authors conclude that there are rhythms in the data? What is the period length of those?

>> EDF 1 has been moved to the main figures and is now Figure 2. We agree with Reviewer 2 that the previous display did not properly reflect the sampling interval. We replaced the continuous line by dots for each sampling point plus a dashed line between the dots.

The presence of a cyclic signal in the opening and closing of mussels was determined with the Dutilleul periodogram analysis that provides the significant periods. We now provide more details on this periodogram, explain why it cannot be plotted, and supplement the figure with the Whittaker-Robinson periodogram for visual purposes. The following text had been added in the Methods section L361-368: "This method estimates significant periodic fractional frequencies and computes the corresponding R-square of the explained variance for each corresponding period. This periodogram is particularly useful when the sampling step does not allow to discover useful periods that are whole multiples of the sampling step. However, plotting the periodogram requires a very large number of fractional values on the abscissa and would be impossible to represent graphically. We thus also show the results of the Whittaker-Robinson-type periodogram that finds a fixed set of whole periods contained in the sampling frequency. The analyses were performed with the R-package *adespatial*."

And the following text has been added in the Results section L98-99: "The Whittaker-Robinson-type periodogram also indicated significant semi-diurnal and diurnal periods (*i.e.* 12 h and its harmonics, p -value < 0.05; Fig. 2c)."

5.) Depending on scientific field the label "in situ" describes something totally different. Please use other words, e.g. "field samples" to label these experiments.

>> We agree with Reviewer 2, the terminology can be specific for each scientific field. Even in the marine biology community at large. Our colleagues working only on littoral environments or species usually make no distinction between "field" and "*in situ*", whereas there is a real distinction for deep-sea researchers between "field work or samples", that designates work performed on board at sea, and "*in situ* work or samples" that designates a work or sample studied in the deep-sea environment, usually with a submersible. We therefore clarify this in the Methods section (L345-347): "In this manuscript, "*in situ*" refers to samples gathered and fixed directly on the seafloor, here at deep-sea hydrothermal vent, to allow a realistic environmental approach."

6.) EDF 7: This EDF is poorly described in the text, albeit pretty important. Move those graphs that show significant rhythms from the field samples to the main figures and describe and depict them well!

Furthermore, in EDF6 the authors show that *B. azoricus* has two *cry1* genes (and proteins). What are the transcript changes for the second *cry1* gene?

>> The Extended Data Figure 7 has been moved to the main Figures and is now Figure 5. We did not separate it into two Figures, one for the *in situ* experiment and one for the lab experiment, because here both experiments convey the message and are important for the comparison with shallow-water mussels. We also describe the Figure in more details (L236-242, plus legend L791).

To identify canonical circadian clock gene candidates in *B. azoricus*, we analysed 3 sources (L516-523): 1) our reference *de novo* transcriptome, 2) a distinct assembly based on the reads (3%) that were not included in the reference transcriptome because their expression was too low to pass the quality thresholds but of interest as single sequences to look for specific candidates, and 3) unpublished *B. azoricus* sequences coming from other sequencing runs (A. Tanguy, co-author). Among the 11 clock candidates we worked on, three of them were provided by A. Tanguy: Baz_3340.1 (NCBI accession number MN611457; one of the two *timeout* homologs), Baz_1996 (NCBI accession number MN611458; one of the two *cry1* homologs), and Baz_1051.1 (NCBI accession number MN611459; *cry DASH*). We used these three sequences to increase the resolution of our phylogenetic trees. As they come from another dataset, we only use the structural information they provide without discussing their expression level. All sequences have been deposited in GenBank. Note that we now mention the NCBI accession numbers to identify our sequences in the trees instead of our "internal" nomenclature.

We clarify all this information in the Method section (L520-523 and L527-529). However, it has to be mentioned that removing the three sequences not coming from the transcriptome would not change the identification of the clock gene candidates, their implication, nor the discussion; it will simply reduce the completeness and resolution of the analysis. Therefore, we are ready to do so if requested.

7.) There are two erratic "?" in the acknowledgement section.

>> We understand this is confusing. The "?" actually belongs to the French name of the oceanographic vessel which is "Pourquoi Pas ?" and to the French name of the research project that co-supported the work which is "Pourquoi pas les abysses ?". We now added " " to these two names to clarify that L800 and L814.

Reviewer #3 (Remarks to the Author):

This is a very interesting manuscript, that expands our knowledge of the biological rhythms to the deep-sea hydrothermal vent invertebrates. In order to support the conclusion, the authors have successfully collected very high-quality behavioural and gene expression data, both of which show biological rhythms with statistical power. In addition, the usage of red light during the *Bathymodiolus* sample collection as well as the deep-sea *in situ* RNA later fixations is highly welcome in deep-sea biology. The RNA-Seq sample preparation and also the downstream statistics are state-of-the-art and stringent. I would thus consider the conclusion very convincing and exciting.

My concern with the experimental part is the laboratory experiment. I understand that the authors would like to find out whether the response to light is an evolutionary relic. However, please note that the deep-sea *Bathymodiolus* mussel has diverged from shallow-water mussel at least 100 million years ago (Lorion et al. 2013; Sun et al. 2017). Light is a non-existing stress/factors in deep-sea *Bathymodiolus* mussel lifetime, and the biological response to light is therefore unreal. The including of this experiment is a bit distracting, and I would suggest the authors remove this part because the rest result of behaviour and *in situ* RNA expression data are collectively very convincing already.

>> Reviewer 3 is right, exposing *B. azoricus* to light is ecologically unrealistic. We do not want to claim that it was, and we now state this very clearly in the manuscript L204 and L274-276. However, we

really believe this experiment has an important added value within this work for several reasons that we set out above, please see the detailed answer to Reviewer 2 major remark 5 (page 7).

We nevertheless understand that this experiment can be considered of less importance and groundbreaking science for the reader. But as explained above, it brings significant inputs to the whole research and to the deep-sea experiment. Therefore, we propose to compromise by moving most of the Figures related to the L:D experiment to the Extended data, and re-designed Figure 3 to emphasise the deep-sea experiment and results. We have organised the text and main Figures accordingly.

In addition, although the overall sampling and experiments are in general well-performed, the manuscript is not very well-written particularly the "Discussion" part, possibly due to the limited words in the current letter format. Notwithstanding, it should be remarked that it is full of thought-provoking observations. In my opinion, from the evolutionary perspective, it would be interesting to compare the result from this study with similar biological rhythms research on shallow-water marine invertebrates, at least with mussels (e.g. Connor & Gracey, 2011), to shed light on the whether the deep-sea *Bathymodiolus* mussel are using similar or dramatically different molecular mechanism in order to find out whether there is a universal tool-kit of biological rhythms.

>> The Discussion has been revised and remodelled. We also moved the text from the Supp Discussion back to the main text (L140-151, L185-190, L279-299). We also revised potential overstatements in the whole manuscript (including not exhaustively L20, L28, L65-72, 279-280). And we re-organised the Discussion section to include a comparison with biological rhythms research on shallow-water mussels in particular (including the work by Connor & Gracey) and put our work in a broader marine chronobiology perspective (L318-L331). Finally, the text was revised by a native English speaker for English corrections.

Minor comments:

One textual issue is the use of 'deep sea' and 'deep-sea'. If you are speaking about a habitat like the deep sea, do not use a hyphen between deep and sea. Please check your complete manuscript. Therefore, remove the hyphen in the Title and also on page 2 line 19, 30, 342, 344, 393 and Figure 1 legend etc.

>> We thank Reviewer 3 for pointing out this mistake. We checked and corrected this throughout the manuscript (L1, 20, 251, 373, 501, 764, 766)

Line 410, 700 g? or 700 mg? Plus, when the mussels were fixed in situ, did the authors try to break the shell to make sure the RNAlater could fully penetrate and fix the tissue?

>> We thank Reviewer 3 for pointing out the typo. The in-house stabilising solution was prepared with 700 grams (g) ammonium sulphate. This has been corrected L394.

Indeed, because the shells were intentionally slightly cracked open using the ROV arm (see Supp. Movie 2) when they were placed in the box, the RNA stabilising solution rapidly permeated mussel tissues to stabilise and protect cellular RNA (attested by the quality of RNA extracts), avoiding the need to immediately process tissue samples. On board, among the 7 to 15 mussels collected by the ROV, we only subsampled mussels that had their shells broken.

As requested by all 3 reviewers, we now provide additional details on the sampling procedures (L 106, L387-392, L395-405, and 425-431) plus videos recorded during the sampling *in situ* (Supp Movies 1 and 2) to be absolutely clear about the sampling protocol. We also provide in-depth clarification to this question in the answers to Reviewer 1.

We thank the 3 Reviewers for having taken the time to review our work carefully and hope our changes respond fully and completely to their relevant comments.

Objet **GenBank MN611450-MN611459**
De <gb-admin@ncbi.nlm.nih.gov>
À <audrey.mat@hotmail.com>, <audrey.mat@ifremer.fr>
Date 2019-10-28 10:55

Dear GenBank Submitter:

Thank you for your direct submission of sequence data to GenBank. We have provided GenBank accession numbers for your nucleotide sequences:

BankIt2276209	Seq1	MN611450
BankIt2276209	Seq2	MN611451
BankIt2276209	Seq3	MN611452
BankIt2276209	Seq4	MN611453
BankIt2276209	Seq5	MN611454
BankIt2276214	Seq1	MN611455
BankIt2276214	Seq2	MN611456
BankIt2276214	Seq3	MN611457
BankIt2276214	Seq4	MN611458
BankIt2276214	Seq5	MN611459

The GenBank accession numbers should appear in any publication that reports or discusses these data, as it gives the community a unique label with which they may retrieve your data from our on-line servers. You may prepare and submit your manuscript before your accessions are released in GenBank.

Submissions are not automatically deposited into GenBank after being accessioned. Each sequence record is individually examined and processed by the GenBank annotation staff to ensure that it is free of errors or problems.

You have requested that your data are to be held confidential until:

Dec 31, 2027

They will not be released to the public database until this date, or until the data or accession numbers appear in print, whichever is first.

Since the flatfile record is a display format only and is not an editable format of the data, do not make changes directly to a flatfile. For complete information about different methods to update a sequence record, see: <https://www.ncbi.nlm.nih.gov/Genbank/update.html>

Any inquiries about your submission should be sent to gb-admin@ncbi.nlm.nih.gov

For more information about the submission process or the available submission tools, please contact GenBank User Support at info@ncbi.nlm.nih.gov.

Please reply using the current Subject line.

Sincerely,

Mark A. Landree, PhD
Contractor

The GenBank Direct Submission Staff
Bethesda, Maryland USA

gb-admin@ncbi.nlm.nih.gov (for updates/replies to GenBank entries)
info@ncbi.nlm.nih.gov (for general questions regarding GenBank)

Objet **GenBank MN597894**
De <gb-admin@ncbi.nlm.nih.gov>
À <audrey.mat@hotmail.com>, <audrey.mat@ifremer.fr>
Date 2019-10-22 21:14

Dear GenBank Submitter:

Thank you for your direct submission of sequence data to GenBank. We have provided a GenBank accession number for your nucleotide sequence:

BankIt2274364 Seq1 MN597894

The GenBank accession number should appear in any publication that reports or discusses these data, as it gives the community a unique label with which they may retrieve your data from our on-line servers. You may prepare and submit your manuscript before your accession is released in GenBank.

Submissions are not automatically deposited into GenBank after being accessioned. Each sequence record is individually examined and processed by the GenBank annotation staff to ensure that it is free of errors or problems.

You have requested that your data are to be held confidential until:

Dec 31, 2021

They will not be released to the public database until this date, or until the data or accession numbers appear in print, whichever is first.

Since the flatfile record is a display format only and is not an editable format of the data, do not make changes directly to a flatfile. For complete information about different methods to update a sequence record, see: <https://www.ncbi.nlm.nih.gov/Genbank/update.html>

Any inquiries about your submission should be sent to gb-admin@ncbi.nlm.nih.gov

For more information about the submission process or the available submission tools, please contact GenBank User Support at info@ncbi.nlm.nih.gov.

Please reply using the current Subject line.

Sincerely,

Lawrence Chlumsky, Ph. D.
Contractor

The GenBank Direct Submission Staff
Bethesda, Maryland USA

gb-admin@ncbi.nlm.nih.gov (for updates/replies to GenBank entries)
info@ncbi.nlm.nih.gov (for general questions regarding GenBank)

Reviewers' Comments:

Reviewer #1:

Remarks to the Author:

The revised manuscript of Mat et al. has addressed some of the questions raised by the three reviewers, and they have done an OK job of adding requested content. I use ambivalent terms purposely - they could have done much better. The authors now address other studies and have toned down some of the initial claims.

Personally, I would have liked them to go further, especially with comparing their differentially expressed genes to those in the *Mytilus* papers. I also think the lack of discussion of physical tidal data from their study sites should be available, or at least its absence discussed. How can you claim a tidal effect if you do not know there are tides present? I'm sure there are, but a blunt acceptance that the lack of this data is a serious omission that should be taken into account when considering the findings should be added.

There are a number of minor errors outlined below.

However, as I said in my first review, this is an extraordinary study and a major feat of deep sea exploration and discovery and I very much hope it is published. I'll leave it to the editor to decide if the above points are critical.

Points that should be corrected or addressed

Did you measure tide flux or other environmental variable at your study site? You cite papers that show these variables operate at vents, but you do not show data for your two sites. Is it available?

[11]

27 organisation in a deep-sea organism. [11] This is vague and unclear as to what you mean. I'd refer to it as "first in situ observation of diurnal transcription cycles in a deep sea organism"

37. mole-rats, that never see daylight.

Publications on mole rats say 'rarely' or 'not frequently exposed to light' they do not use 'never' The argument that rhythms are 'adaptive' is also vague. Maybe they are in fact exposed to light on occasion, or maybe light is not the entraining factor.

140. What units are these? normalized read counts/dataset, TPM...?

170. Similar questions have been studied in coral- see PMID: 24508015

The supplemental files need titles and captions that make sense. A title such as "235662_1_data_set_4556167_q8555s" make zero sense

183. Units?

243. "profoundly ground-breaking " is a split infinitive, just say 'ground-breaking'

263. I'm not sure what this means 'our understanding of ecosystem dynamics could be dramatically transformed and be even fundamental in estimating biodiversity '

298. There are many studies in corals and the marine midge

309. Although the authors do discuss the work on *Mytilus* in sufficient depth. How many DEG are in common, what cycles do these follow? they do not delve very deeply, and no intersection of genes found in the various studies is here.

[11]

[SEP]

[11]

[SEP]

Reviewer #2:

Remarks to the Author:

The authors have successfully addressed our major and minor comments.

There are three points remaining that should be addressed:

Lines 41-43: The word "loss" is probably incorrect. See Sci Rep. 2018 Sep 27;8(1):14466. doi: 10.1038/s41598-018-32778-4. Better talk of a reduction.

L:D lab experiment:

Lines 177 onwards: The new motivation of the LD experiment described in the text is very helpful, as are the interpretations from line 204 onwards.

Remaining question- as the experiment is motivated to understand if the mussels respond to light then providing information on the exact spectrum and intensity of the light used is important. (Provide spectrum and intensity in photons /area*time unit*wavelength.) Just providing the lamp information is not sufficient, because especially intensity strongly depends on the distance to the sample.

This information is important to interpret if the light exposure is at all physiological and if yes, what kind of photoreceptor(s) might be underlying the response.

Concerning the authors' offer to remove the tim, cry1 and cry dash sequences not coming from the current transcriptome: The decision what to do should depend on how sure the authors are that these are truly paralogous genes and not caused by many polymorphisms in a single cry1/ tim/ cry-dash gene. Could it be cases of sequence differences in single genes due to cryptic speciation?

If there is doubt, the recommendation is to remove them, otherwise they should remain in the manuscript, because they provide quite interesting additional information.

Reviewer #3:

Remarks to the Author:

Dear Colleagues,

Thank you for responding.

In general, I am happy with the revisions. Intrigued by your findings, I am also thinking the Bathymodiolus mussel could sense the light emitted from deep-sea hydrothermal vent (see the work performed by White et al., 2002). Although it is weak, the vent field is not completely dark. Other than that, I don't have any other comments. Great work!

Best regards,

Pei-Yuan Qian

Reviewers' comments:

Reviewer #1 (Remarks to the Author):

The revised manuscript of Mat et al. has addressed some of the questions raised by the three reviewers, and they have done an OK job of adding requested content. I use ambivalent terms purposely - they could have done much better. The authors now address other studies and have toned down some of the initial claims.

Personally, I would have liked them to go further, especially with comparing their differentially expressed genes to those in the *Mytilus* papers. I also think the lack of discussion of physical tidal data from their study sites should be available, or at least its absence discussed. How can you claim a tidal effect if you do not know there are tides present? I'm sure there are, but a blunt acceptance that the lack of this data is a serious omission that should be taken into account when considering the findings should be added.

>> We address the point of the comparison with *Mytilus* species in the extended question below.

We agree with Reviewer 1, physical tidal data are a crucial piece of information for the context of the present work. We actually provided *in situ* pressure and temperature data and associated periodogram analyses that highlighted a tidal variation in environmental conditions at the study site, at the time of sampling. These data are mentioned in the Result section (L135-141) and in Figure 3c. Temperature and pressure were measured at the Lucky Strike vent field using a probe deployed on the EMSO infrastructure (Methods, L384-386). We understand from Reviewer 1's comment that this was not clear enough, and we now provide the precise location of data acquisition by giving the distance between the probe and our sampling site (745 m; L386).

The predominant role of tides in controlling fluid flux direction and intensity at hydrothermal vents has been intensively studied, as already mentioned in the Introduction (L56-61). The influence of tides at vents results from oceanic tidal pressure (Crone & Wilcock 2005) and the modulation of bottom currents by ocean tides (Little *et al.* 1988; Tivey *et al.* 2002; Scheirer *et al.* 2006; Barreyre *et al.* 2014). These mechanisms have been shown to impact habitat conditions in all oceans including the Atlantic (Kinoshita *et al.* 1998; Barreyre *et al.* 2014; Sarrazin *et al.* 2014; Cuvelier *et al.* 2017) and the Pacific (Johnson & Tunnicliffe 1985; Chevaldonné *et al.* 1991; Schultz *et al.* 1992; Johnson *et al.* 1994; Lee *et al.* 2015; Lelièvre *et al.* 2017) with an impact on organisms growth (Schöne & Giere 2005; Nedoncelle *et al.* 2015). Tidal variations also occur in end-member (focus) fluid in terms of temperature and chemistry (Larson *et al.* 2007, Barreyre *et al.* 2014) and plume motion (Xu & Di Iorio 2012). The challenge is now to understand how this tide-related variability in temperature, chemistry and currents affect species biology, ecology and behaviour.

At Lucky Strike, the deployment of temperature probes across a high number of sites (including Tour Eiffel, our study site) revealed that high-temperature discharge correlates to tidal pressure while low-temperature discharge correlates to tidal currents (Barreyre *et al.* 2014). In addition, temperature measurements performed in mussel habitats on our study site, also showed a strong tidal variability in temperature and iron concentrations (Sarrazin *et al.* 2014, Cuvelier *et al.* 2017). This information is now better emphasised in the discussion (L264): "In situ pressure and temperature data recorded at LS at the time of sampling confirmed the prominent role of tides on environmental variability, as previously shown in both focus flow and mussel habitats at the same site^{13,16}."

In the light of the environmental data provided in the paper and previous knowledge from Lucky Strike, and vents in general, we believe that the presence of tides is undeniable. Others articles exists establishing the role of tides on vent systems (e.g. Crone & Wilcock 2005; Scheirer *et al.* 2006; Tivey *et al.* 2002), but the limit of the number of references is reached, and we believe that our changes, thanks to Reviewer 1, are now enough to support tidal signals.

1. Barreyre, T. *et al.* Temporal variability and tidal modulation of hydrothermal exit-fluid temperatures at the Lucky Strike deep-sea vent field, Mid-Atlantic Ridge: MAR vent-field temperature monitoring. *Journal of Geophysical Research: Solid Earth* **119**, 2543–2566 (2014).
2. Chevalloné, P., Desbruyères, D. & Haïre, M. L. Time-series of temperature from three deep-sea hydrothermal vent sites. *Deep Sea Research Part A. Oceanographic Research Papers* **38**, 1417–1430 (1991).
3. Crone, T. J. & Wilcock, W. S. D. Modeling the effects of tidal loading on mid-ocean ridge hydrothermal systems: HYDROTHERMAL SYSTEMS. *Geochem. Geophys. Geosyst.* **6**, n/a-n/a (2005).
4. Cuvelier, D., Legendre, P., Laës-Huon, A., Sarradin, P.-M. & Sarrazin, J. Biological and environmental rhythms in (dark) deep-sea hydrothermal ecosystems. *Biogeosciences* **14**, 2955–2977 (2017).
5. Johnson, K. S., Childress, J. J., Beehler, C. L. & Sakamoto, C. M. Biogeochemistry of hydrothermal vent mussel communities: the deep-sea analogue to the intertidal zone. *Deep Sea Research Part I: Oceanographic Research Papers* **41**, 993–1011 (1994).
6. Johnson, H. P. & Tunncliffe, V. Time-series measurements of hydrothermal activity on northern Juan De Fuca Ridge. *Geophys. Res. Lett.* **12**, 685–688 (1985).
7. Kinoshita, M., Von Herzen, R. P., Matsubayashi, O. & Fujioka, K. Tidally-driven effluent detected by long-term temperature monitoring at the TAG hydrothermal mound, Mid-Atlantic Ridge. *Physics of the Earth and Planetary Interiors* **108**, 143–154 (1998).
8. Larson, B. I., Olson, E. J. & Lilley, M. D. In situ measurement of dissolved chloride in high temperature hydrothermal fluids. *Geochimica et Cosmochimica Acta* **71**, 2510–2523 (2007).
9. Lee, R. W., Robert, K., Matabos, M., Bates, A. E. & Juniper, S. K. Temporal and spatial variation in temperature experienced by macrofauna at Main Endeavour hydrothermal vent field. *Deep Sea Research Part I: Oceanographic Research Papers* **106**, 154–166 (2015).
10. Lelièvre, Y. *et al.* Astronomical and atmospheric impacts on deep-sea hydrothermal vent invertebrates. *Proceedings of the Royal Society B: Biological Sciences* **284**, 20162123 (2017).
11. Little, S. A., Stolzenbach, K. D. & Grassle, F. J. Tidal current effects on temperature in diffuse hydrothermal flow: Guaymas Basin. *Geophys. Res. Lett.* **15**, 1491–1494 (1988).
12. Nedoncelle, K. *et al.* Bathymodiolus growth dynamics in relation to environmental fluctuations in vent habitats. *Deep Sea Research Part I: Oceanographic Research Papers* **106**, 183–193 (2015).
13. Sarrazin, J., Cuvelier, D., Peton, L., Legendre, P. & Sarradin, P. M. High-resolution dynamics of a deep-sea hydrothermal mussel assemblage monitored by the EMSO-Açores MoMAR observatory. *Deep Sea Research Part I: Oceanographic Research Papers* **90**, 62–75 (2014).
14. Scheirer, D. S., Shank, T. M. & Fornari, D. J. Temperature variations at diffuse and focused flow hydrothermal vent sites along the northern East Pacific Rise: EPR VENT TEMPERATURES. *Geochemistry, Geophysics, Geosystems* **7**, n/a-n/a (2006).
15. Schöne, B. R. & Giere, O. Growth increments and stable isotope variation in shells of the deep-sea hydrothermal vent bivalve mollusk *Bathymodiolus brevior* from the North Fiji Basin, Pacific Ocean. *Deep Sea Research Part I: Oceanographic Research Papers* **52**, 1896–1910 (2005).
16. Schultz, A., Delaney, J. R. & McDuff, R. E. On the partitioning of heat flux between diffuse and point source seafloor venting. *J. Geophys. Res.* **97**, 12299 (1992).
17. Tivey, M. K., Bradley, A. M., Joyce, T. M. & Kadko, D. Insights into tide-related variability at seafloor hydrothermal vents from time-series temperature measurements. *Earth and Planetary Science Letters* **202**, 693–707 (2002).
18. Xu, G. & Di Iorio, D. Deep sea hydrothermal plumes and their interaction with oscillatory flows: HYDROTHERMAL PLUMES IN OSCILLATORY FLOWS. *Geochem. Geophys. Geosyst.* **13**, (2012).

There are a number of minor errors outlined below.

However, as I said in my first review, this is an extraordinary study and a major feat of deep sea exploration and discovery and I very much hope it is published. I'll leave it to the editor to decide if the above points are critical.

Points that should be corrected or addressed

Did you measure tide flux or other environmental variable at your study site? You cite papers that show these variables operate at vents, but you do not show data for your two sites. Is it available?

>> Yes, we measured and provide environmental data showing a tidal signal *in situ* at Lucky Strike (see above). Mussels samples at Menez Gwen where further studied in the lab under a 12:12 L:D schedule without tides. We did not measure variables over time at Menez Gwen and there is no data showing environmental variability at that site, but as mentioned in our response to the first point above, the role of tides on vent conditions through bottom currents modulation and tidal pressure is widely recognised and is now a given fact in the deep-sea community. A general model of mechanisms involved is available for hydrothermal system on mid-ocean ridges (Crone & Wilcock, 2005).

We added a sentence in the discussion to stress the fact that we acquired tidal data but also cited a paper that report the influence of tides at Lucky Strike (L264). We choose this paper (Barreyre *et al.* 2014) because it was already cited elsewhere in the article.

27 organisation in a deep-sea organism. This is vague and unclear as to what you mean. I'd refer to it as "first in situ observation of diurnal transcription cycles in a deep sea organism"

>> We understand the comment from Reviewer 1; however, we provide not only transcriptional data but also behavioural data. Also, these data show daily cycles but also tidal cycles. This is why we refer to the "temporal organisation" of the mussel. Including the whole range of observations would result in a long complicated sentence, largely exceeding the number of allowed characters in the Abstract. As the meaning of what the "temporal organisation" involves and refers to is detailed just above, we hope this is clear enough for the reader.

37. mole-rats, that never see daylight.

Publications on mole rats say 'rarely' or 'not frequently exposed to light' they do not use 'never'

The argument that rhythms are 'adaptive' is also vague. Maybe they are in fact exposed to light on occasion, or maybe light is not the entraining factor.

>> We changed "never" and wrote "rarely" instead (L37). It is indeed difficult to know how often mole-rats see daylight. Oosthuizen *et al.* (2003) mention "hardly, if ever", but we agree with Reviewer 1, it might be incorrect to state that it is the case over the whole lifetime of the animal. And there might indeed be other entraining factors.

Showing rigorous evidences that rhythms are adaptive is a central question and challenge in chronobiology, and several authors have addressed this question (e.g. Yerushalmi and Green, 2009 *Ecol Lett* 12:970-981; Nikhil and Sharma, 2017, Chapter 5 in Kumar V. (eds) *Biological Timekeeping: Clocks, Rhythms and Behaviour*). Biological rhythms in species living in aperiodic environments have been one of the arguments used to suggest that biological rhythms are adaptive. We do not exhaustively mention all these studies, but we hope that the correction we now make and that the precautions we take in the introduction by using "hypothesis that internal timing is probably adaptive" (L38) and "may be of ecological value" (L41) will appear as a fair assessment of the current knowledge in the field.

140. What units are these? normalized read counts/dataset, TPM...?

>> We thank Reviewer 1 for this comment, we now mention that these are normalised counts, a standard commonly used in RNAseq articles (L150, L194, L248).

170. Similar questions have been studied in coral- see PMID: 24508015

>> We agree, this question is not only relevant to deep-sea mussels. We now added a reference in the text to the article from Sorek *et al.* (2018) on sea anemones (L181). That work as fully relevant and was already referenced in the manuscript. Unfortunately, we cannot exhaustively mention all studies addressing the question of how biological rhythms work in a symbiotic interaction as we are already in the upper limit of the number of references allowed by the Journal, but we hope that this will show the broader relevance of the question to the reader.

The supplemental files need titles and captions that make sense. A title such as "235662_1_data_set_4556167_q8555s" make zero sense

>> As mention in our email to the Editor, we do not understand to what data set this name refers: all the files we provided were named using "Mat_et_al" followed by either figure or data or movie and their number. It appears that it is an issue related to the journal automatic renaming and not coming from our naming code.

183. Units?

>> We now mention that these are normalised counts (L150 and 194).

243. "profoundly ground-breaking " is a split infinitive, just say 'ground-breaking'
>> This has been modified L258.

263. I'm not sure what this means 'our understanding of ecosystem dynamics could be dramatically transformed and be even fundamental in estimating biodiversity '

>> Biological rhythms influence organisms' temporal niche, *i.e.* the 24h temporal distribution of physiological and behavioural processes. Mussels are semi-sessile animals. However, other species can move, and therefore be present or not in a specific area at a given time depending on their biological cycle. Taking into account potential biological rhythms influence in sampling strategies could therefore be crucial to determine biodiversity in deep-sea ecosystems. As an example, sampling in tidal flats always includes timing in the planning; indeed, observations made at high or low tide will lead to different biodiversity patterns. We modified the sentence because its structure may have led to confusion (L286): "*Including biological rhythms and temporal niches into future research protocols and environmental protection strategies could fundamentally transform our understanding of ecosystem functioning and biodiversity patterns.*"

298. There are many studies in corals and the marine midge

>> Reviewer 1 is right, there are indeed studies in corals and the marine midge, as well as in crustaceans, oysters, marine worms, and others. We actually mention several studies related to these organisms (L33-35, L127-128, L327-332), but it will be difficult to reference them all. However, what we meant to highlight is the huge gap in terms of both number of studies and knowledge gathered in chronobiology when we compare marine and terrestrial species (L326-327): "*Knowledge on molecular biological clock(s) in marine species is severely lacking and only a few studies exist compared to data on terrestrial organisms.*"

Biological rhythms are way more studied in terrestrial species than marine ones. Similarly, molecular studies of biological rhythms in marine species are recent. They have mostly been acquired during the last decade, including in corals and the marine midge, while the cloning of the first circadian clock gene in *Drosophila* was performed in the 1980s (Reddy et al, 1984, Cell 38:701-710; Bargiello et al, 1984, Nature 312:752-754). The molecular circatidal clock is today still uncharacterised (L33-36). The connection between the circadian and circatidal clocks is an open question (L332-334). The first insights into that question were provided in 2013, as mentioned L327. For a while, one of the reason was the absence a genetic model systems in marine species. Several teams are now doing a great work to providing such models, and we acknowledge their work (L327-330). Genomic technologies also opened the doors to broader studies in non-model organisms. Nevertheless, we maintain that the gap between the knowledge we have in terrestrial chronobiology and marine chronobiology is important, as stated by other researchers (among others Bulla et al, 2017, Phil. Trans. R. Soc. B 372: 20160253; de la Iglesia and Johnson, 2013, Curr Biol 23(20): R921). Deep-sea chronobiology is thus an exciting emerging field of research.

309. Although the authors do discuss the work on *Mytilus* in sufficient depth. How many DEG are in common, what cycles do these follow? they do not delve very deeply, and no intersection of genes found in the various studies is here.

>> We misunderstood the question of Reviewer 1 during the first round of review as a request to compare rhythms in shallow- and deep-water mussels in general, as detailed L336-346.

We now better understand Reviewer 1's request. It is a great suggestion: we now consider the physiology of deep- and shallow water mussels at the Gene Ontology term level (L275-282).

We believe that the comparison is currently more relevant at the Gene Ontology term level rather than at the individual gene level. First, it has been reported in mussels (Connor and Gracey, 2011, PNAS 108(38): 16110-16115) that there is a variability in the identification of specific rhythmic transcripts between experiments. Working at the GO term level is therefore more robust. Second, studies were performed with different techniques and algorithms. For example, Gracey et al., 2008 (Curr Biol 18: 1501-1507) used singular-value decomposition to identify the main patterns of temporal gene

expression. GO terms are provided. In Connor and Gracey, 2011, the list of all the tidal and daily transcripts are not provided, but they provide specific subsets. GO biological process categories associated with low or high tide are provided. So, GO term comparison appears as the most relevant way to consider how biological rhythms influence shallow- and deep-water mussel physiology, and we now provide this comparison (L275-282).

Reviewer #2 (Remarks to the Author):

The authors have successfully addressed our major and minor comments.
There are three points remaining that should be addressed:

Lines 41-43: The word "loss" is probably incorrect. See Sci Rep. 2018 Sep 27;8(1):14466. doi: 10.1038/s41598-018-32778-4. Better talk of a reduction.

>> "Loss" has been replaced by "attenuation" because "reduction" is already used further in the sentence (L40).

L:D lab experiment:

Lines 177 onwards: The new motivation of the LD experiment described in the text is very helpful, as are the interpretations from line 204 onwards.

Remaining question- as the experiment is motivated to understand if the mussels respond to light then providing information on the exact spectrum and intensity of the light used is important. (Provide spectrum and intensity in photons /area*time unit*wavelength.) Just providing the lamp information is not sufficient, because especially intensity strongly depends on the distance to the sample.

This information is important to interpret if the light exposure is at all physiological and if yes, what kind of photoreceptor(s) might be underlying the response.

>> We added the information about the lighting condition for the lab experiment L430-432. The spectrum is provided as colour temperature (in Kelvin), and the intensity is provided as lux measured in the air both at water surface and at the bottom of the empty aquarium. These are the information we could gather and measure for the experiment.

While the information might help understanding what kind of photoreceptor(s) might be involved in the response, we used white light in the lab, so further experiments will be necessary to dig into the details of photoreceptors and specific wavelength sensitivity.

Concerning the authors' offer to remove the *tim*, *cry1* and *cry dash* sequences not coming from the current transcriptome: The decision what to do should depend on how sure the authors are that these are truly paralogous genes and not caused by many polymorphisms in a single *cry1/ tim/ cry-dash* gene. Could it be cases of sequence differences in single genes due to cryptic speciation?

If there is doubt, the recommendation is to remove them, otherwise they should remain in the manuscript, because they provide quite interesting additional information.

>> We did an important effort to synthesise the sequences we identified by BLAST to keep only the most relevant ones while providing the most complete information. We did not keep clearly redundant sequences. For the remaining sequences: without a genome for *B. azoricus*, we cannot be totally sure whether the *tim* and *cry1* sequences are isoforms or paralogous genes because these sequences do not totally overlap and because we are lacking genomic information about this locus. However, even in the case where these sequences are isoforms and not paralogous genes, it has been reported that several isoforms can have a functional significance. For example, this could point towards an interesting possibility of cross-regulation via reciprocal inhibition, as it is known for other proteins involved in developmental timing in insects (Thummel, 1997, BioEssays 19(8): 669; Mane-Padros et al., 2005, Dev Biol 315: 147-160). In this case, this information is of biological importance. Genetic and mechanistic variability can arise from isoforms as well as from gene duplication, although it is usually more complicated to study for isoforms. Concerning cryptic speciation: this is indeed a possibility, but it this stage not the most probable one.

In the absence of a genome for *B. azoricus*, we tried to be as concise but also as complete as possible. We believe that all these sequences are worth mentioning and deserve to be digged into for future research in deep-sea chronobiology. However, to avoid any over-interpretation of our data, we removed the word "homologs" L231.

Reviewer #3 (Remarks to the Author):

Dear Colleagues,

Thank you for responding.

In general, I am happy with the revisions. Intrigued by your findings, I am also thinking the Bathymodiolus mussel could sense the light emitted from deep-sea hydrothermal vent (see the work performed by White et al., 2002). Although it is weak, the vent field is not completely dark. Other than that, I don't have any other comments. Great work!

Best regards,

Pei-Yuan Qian

>> Professor Qian is right, deep-sea hydrothermal vents are blind to the light:dark solar cycle, but they emit low levels of thermal and non-thermal radiation. We now mention this in the Discussion (L293-294) to be as complete as currently possible on that question.

We thank again the 3 Reviewers for having taken the time to review our work. We feel that after addressing these constructive comments, the manuscript is strengthened and has gained in clarity. We really hope our changes clarify the remaining points.